# Adolescent stress remodels synapses in the sensory thalamus and impairs tactile discrimination in mice
Hisako Nakayama ⬡ ✉ & Mariko Miyata ⬡ ✉

Adolescent stress can lead to persistent behavioral issues and brain dysfunction, often linked to sensory perception abnormalities. The biological basis for these sensory changes is not well understood. Here, we demonstrate that social isolation during adolescence, but not adulthood, remodels the structure and function of brain circuits that process touch in mice. Specifically, isolation increases the number of afferents to each neuron while weakening individual connections, and these changes depend on the activation of glucocorticoid receptors in thalamic neurons. Mice exposed to adolescent isolation also show reduced ability to distinguish fine tactile differences, suggesting a functional impact of the altered thalamic circuitry. Cohabitation with another mouse can partially reverse the synaptic effects of adolescent isolation. Our findings identify the sensory thalamus as a vulnerable site for adolescent stress and show how social experiences during this period can remodel sensory circuits, providing new directions for preventing or treating stress-related sensory problems.

Psychological stress is an inevitable part of daily life. Severe and chronic psychological stress is thought to be a risk factor for a wide range of psychosomatic and mental disorders[1–3]. In particular, psychological stress during childhood and adolescence exerts profound effects, as adverse experiences such as abuse and neglect during these develpmental periods can lead to emotional disorders and substance addiction in adulthood[4–6]. Psychological stress also affects sensory systems. For example, psychological stress can induce sudden hearing loss[7,8] and exacerbate itching[9,10], both of which are increasingly recognized as stress-related conditions in modern society. Moreover, adolescents under social exclusion exhibit heightened sensitivity to thermal stimuli[11], and unusual sensory profiles are correlated with traumatic experiences during childhood[12]. These atypical sensory experiences can themselves harm the mental state and reduce the quality of life[12–16]. Therefore, understanding how psychological stress, especially during adolescence, alters sensory processing at the neuronal level is crucial for developing therapeutic strategies to prevent or ameliorate these sensory abnormalities. In this study, we aimed to elucidate the neural basis for adolescent stress-induced sensory alterations at the synaptic level.

We used social isolation (SI) mice, an experimental model that mimics human social isolation by housing animals individually and restricting social contact. Using this model, we investigated how adolescent SI affects synaptic transmission and connectivity, which relay tactile inputs to whiskers, and how these changes influence the whisker-related behaviors in mice. Tactile inputs to whiskers, a key sensory input in mice, are transmitted from the brainstem maxillary principal trigeminal nucleus (PrV2) to the ventral posterior medial nucleus of the thalamus (VPM), which then sends it to the primary somatosensory cortex (S1)[17,18]. Early in postnatal development, multiple medial lemniscus fibers (MLFs) containing axons from PrV2 neurons and non-PrV2 neurons form weak excitatory synapses on each VPM neuron[19–21]. By postnatal day 21 (P21), only one PrV2 neuron among the MLFs to each VPM neuron is selectively maintained, while synapses by other MLFs are eliminated[19–21]. Sensory experience with the whiskers during this period is necessary for this refinement, since whisker deprivation at P12-P13 impairs the elimination of MLFs and keeps multiple, redundant MLFs inputs beyond P21[19,22]. However, little is known about how these refined synaptic connections are maintained after weaning or what factors might destabilize them later in development.

We found that adolescent SI significantly increased the number of MLFs forming synapses to each VPM neuron while weakening the strength of each MLF. Mice experienced adolescent SI displayed deficits in whisker-related behaviors. The remodeling of MLF-VPM synapses during adolescent SI is mediated by glucocorticoid receptors (GRs) in VPM neurons, activated by transient elevation of corticosterone levels during the isolation period. We also found that synaptic remodeling caused by adolescent SI is partly restored through cohabitation with another mouse during adolescence. Together, our study demonstrates that adolescence is a crucial period during which vulnerability to SI increases, and stress-induced GR

Division of Neurophysiology, Department of Physiology, School of Medicine, Tokyo Women's Medical University, Shinjuku-ku, Tokyo, Japan.
✉e-mail: nakayama.hisako@twmu.ac.jp; mmiyata@twmu.ac.jp

overactivation can alter thalamic neural circuits, thereby disrupting sensory processing.

## Results

### Adolescent social isolation remodeled MLF-VPM synapses

We primarily use male mice in most experiments. In the weaning period (P20–P22), 68% of VPM neurons were innervated by a single MLF and the mean number of afferents was $1.50 \pm 0.13$ (n = 44 cells, N = 3 mice, Supplementary Fig. 1c). The single-fiber amplitude (SF-amplitude) and the maximum amplitude (Max-amplitude) in each VPM neuron were $2468 \pm 229$ pA (n = 66 fibers, N = 3 mice) and $3713 \pm 264$ pA (n = 44 cells, N = 3 mice), respectively, as shown previously[19,20] (Supplementary Fig. 1a–c). After four weeks of group-housed (GH) or socially isolated (SI) rearing from P21, electrophysiological experiments were conducted (Fig. 1a). The number of afferents was significantly greater in SI mice than GH mice (GH, $1.35 \pm 0.08$, n = 49, N = 4; SI, $2.27 \pm 0.15$, n = 49, N = 5; $p < 0.001$ by $U$-test; Fig. 1b, c). SF-amplitude in SI mice was significantly smaller than GH mice, while no difference was observed in Max-amplitude between SI and GH mice (SF-amplitude: GH, $2126 \pm 184$ pA, n = 66, N = 4; SI, $1155 \pm 86$ pA, n = 108, N = 5; $p < 0.001$ by $U$-test; Max-amplitude: GH, $2863 \pm 251$ pA, n = 49, N = 4; SI, $2517 \pm 189$ pA, n = 49, N = 5; $p = 0.637$ by $U$-test; Fig. 1b,d). The paired-pulse ratio (PPR) did not change by SI rearing (GH, $0.66 \pm 0.01$, n = 49, N = 4; SI, $0.67 \pm 0.01$, n = 49, N = 5; $p = 0.305$ by $U$-test; Supplementary Fig. 2a), suggesting that the presynaptic release machinery does not differ between GH mice and SI mice. These findings showed that the adolescent SI rearing significantly altered MLF-VPM synapses.

We then investigated whether the susceptibility of MLF-VPM synapses to SI rearing is limited to a specific life stage. Mice with a four-week SI starting at P35 showed a significant increase in afferent number and a decrease in SF-amplitude compared to GH mice as was the case for SI mice starting at P21 shown in Fig. 1a–d (afferent number: GH, $1.49 \pm 0.11$, n = 37, N = 3; SI, $2.14 \pm 0.15$, n = 37, N = 3; $p < 0.001$ by $U$-test; SF-amplitude: GH, $1899 \pm 189$ pA, n = 55, N = 3; SI, $1228 \pm 161$ pA, n = 79, N = 3; $p < 0.001$ by $U$-test; Supplementary Fig. 3). Next mice were reared in SI or GH conditions from P56, when the mice were sexually mature adults. The adult SI mice did not show a difference in afferent number (GH, $1.46 \pm 0.09$, n = 52, N = 4; SI, $1.44 \pm 0.12$, n = 45, N = 4; $p = 0.544$ by $U$-test; Fig. 1e–h). In this condition, both SF- and Max-amplitudes were significantly smaller in SI than in GH mice (SF-amplitude: GH, $2132 \pm 172$ pA, n = 75; N = 4; SI, $1410 \pm 136$ pA, n = 65, N = 4; $p < 0.001$ by $U$-test; Max-amplitude: GH, $3075 \pm 199$ pA, n = 52, N = 4; SI, $2037 \pm 179$ pA, n = 45, N = 4; $p < 0.001$ by $U$-test; Fig. 1h). There was no difference in PPR (GH, $0.66 \pm 0.01$, n = 52, N = 4; SI, $0.67 \pm 0.01$, n = 45, N = 4; $p = 0.548$ by $U$-test; Supplementary Fig. 2b). These results showed that the four-week SI rearing attenuates synaptic transmission at each MLF afferent regardless of age but induces synapse formation from new MLF afferents only during adolescence. The reduction in Max-amplitude in adult SI mice may correspond to the lack of an increase in afferent fibers.

### Adolescent SI induces remodeling of MLF-VPM synapses in females, as it does in males

We tested sex differences in the responsiveness of MLF-VPM synapses to adolescent and adult SI. When female mice were reared in GH or SI conditions for four weeks from P21, the afferent number was significantly larger and SF-amplitude was smaller in adolescent SI than in GH mice (afferent number: GH, $1.43 \pm 0.12$, n = 28, N = 3; SI, $2.40 \pm 0.20$, n = 35, N = 3; $p < 0.001$ by $U$-test; SF-amplitude: GH, $1798 \pm 197$ pA, n = 40, N = 3; SI, $1008 \pm 88$ pA, n = 85, N = 3; $p < 0.001$ by $U$-test; Supplementary Fig. 4a–d). Max-amplitude and PPR were not different between the two conditions (Max-amplitude: GH, $2569 \pm 246$ pA, n = 28, N = 3; SI, $2449 \pm 214$ pA, n = 35, N = 3; $p = 0.701$ by $U$-test; PPR: GH, $0.67 \pm 0.02$, n = 28, N = 3; SI, $0.66 \pm 0.01$, n = 35, N = 3; $p = 0.422$ by $U$-test; Supplementary Figs. 2c, 4c). When female mice were reared in GH or SI conditions from P58, afferent number and PPR did not differ between rearing conditions (afferent

number: GH, $1.47 \pm 0.11$, n = 55, N = 4; SI, $1.74 \pm 0.12$, n = 57, N = 4; $p < 0.053$ by $U$-test; PPR: GH, $0.74 \pm 0.01$, n = 55, N = 4; SI, $0.74 \pm 0.01$, n = 57, N = 4; $p = 0.417$ by $U$-test; Supplementary Figs. 2d, 4e–g). SF- and Max-amplitudes were significantly smaller in SI than in GH mice (SF-amplitude: GH, $1806 \pm 134$ pA, n = 81, N = 4; SI, $1216 \pm 92$ pA, n = 99, N = 4; $p < 0.001$ by $U$-test; Max-amplitude: GH, $2660 \pm 153$ pA, n = 55, N = 4; SI, $2112 \pm 146$ pA, n = 57, N = 4; $p = 0.010$ by $U$-test; Supplementary Fig. 4h). These data indicate that changes in MLF-VPM synapses after SI rearing are similar between females and males. When parameters of MLF-VPM synapses are directly compared between males and females reared in the same conditions, we found that the afferent number was slightly, but significantly, larger in adult SI females than in adult SI males ($p = 0.039$ by $U$-test). This suggests that the direction and extent of changes in MLF-VPM synapses under SI are similar between sexes, but synapse formation tended to occur more readily in females than in males during adulthood.

### Adolescent SI rearing does not increase synapses from the non-whisker brainstem regions

We previously reported that multiple innervations of VPM neurons at MLF-VPM synapses reappeared after cutting the peripheral infraorbital nerve (IONC)[23], and that the newly formed synapses were derived from afferents outside the brainstem PrV2 region[24]. To examine whether adolescent SI promotes the formation of synapses from areas other than PrV2, we performed histological analyses using Krox20-Ai34D transgenic mice and immunostaining for VGluT2[21,24–26]. Since all excitatory synapses ascending from the brainstem express VGluT2, and among them, those from PrV2 express tdTomato in Krox20-Ai34D mice, this enabled us to distinguish synapses from PrV2 and non-PrV2 as VGluT2(+)/tdTomato(+) puncta and VGluT2(+)/tdTomato(−) puncta, respectively (Supplementary Fig. 5a–h)[24–26]. Adolescent SI did not alter the density of VGluT2(+) puncta in the VPM (GH, $15.8 \pm 1.04 \times 10^{-3}/\mu m^2$, N = 6; SI, $16.9 \pm 0.57 \times 10^{-3}/\mu m^2$, N = 6; $p = 0.378$ by Student's $t$-test; Supplementary Fig. 5i), corresponding to the electrophysiological data showing that the Max-amplitude was not altered by adolescent SI (Fig. 1d). Unlike after IONC, the ratio of VGluT2(+)/tdTomato(−) puncta to VGluT2(+) puncta did not differ between GH and SI conditions (GH, $0.27 \pm 0.02$, N = 6; SI, $0.25 \pm 0.02$, N = 6; $p = 0.488$ by Student's $t$-test; Supplementary Fig. 5j). These results suggest that adolescent SI promotes the formation of synapses from PrV2 rather than from non-PrV2 regions.

### Adolescent SI impairs whisker-related behaviors

We then analyzed how tactile sensation mediated by the whiskers is affected by adolescent and adult SI. To evaluate the tactile response to the whiskers, we examined the response threshold of mice to mechanical stimulation of the whisker pad using von Frey filaments, following established methods[24,26–28] (Supplementary Fig. 6b, e). The response threshold of adolescent SI mice was significantly lower than that of adolescent GH mice (SI, $0.71 \pm 0.20$ g, N = 12; GH, $3.90 \pm 0.84$ g, N = 12; $p = 0.004$ by $U$-test; Supplementary Fig. 6b). In contrast, the response thresholds did not differ between adult GH and SI mice (SI, $3.60 \pm 0.50$ g, N = 10; GH, $3.06 \pm 0.50$ g, N = 10; $p = 0.462$ by $U$-test; Supplementary Fig. 6e). These results suggest that tactile thresholds are reduced by adolescent SI rearing. However, since the stimuli were applied while the mice were held by the experimenter, we cannot rule out the possibility that the restraint influenced the response threshold. Therefore, we examined how whisker-using behaviors under unrestrained conditions are affected by SI using two additional behavioral paradigms: the tactile novel object recognition test (tNORT, Fig. 2)[29–32] and the gap-cross test (Supplementary Fig. 6c, f)[33,34].

A series of tNORT experiments consisted of an open field test (OFT) on day 1, tNORT on day 2 and 3, and the novel object recognition test (NORT) on day 4 (Fig. 2). OFT revealed no difference in total distance traveled or center time, indicating that locomotor activity and anxiety levels in the testing arena did not differ among adolescent and adult GH and SI mice (Total distance: Adolescence-GH, $33.8 \pm 1.08$ m, N = 16; SI, $36.2 \pm 1.28$ m,

**Article**

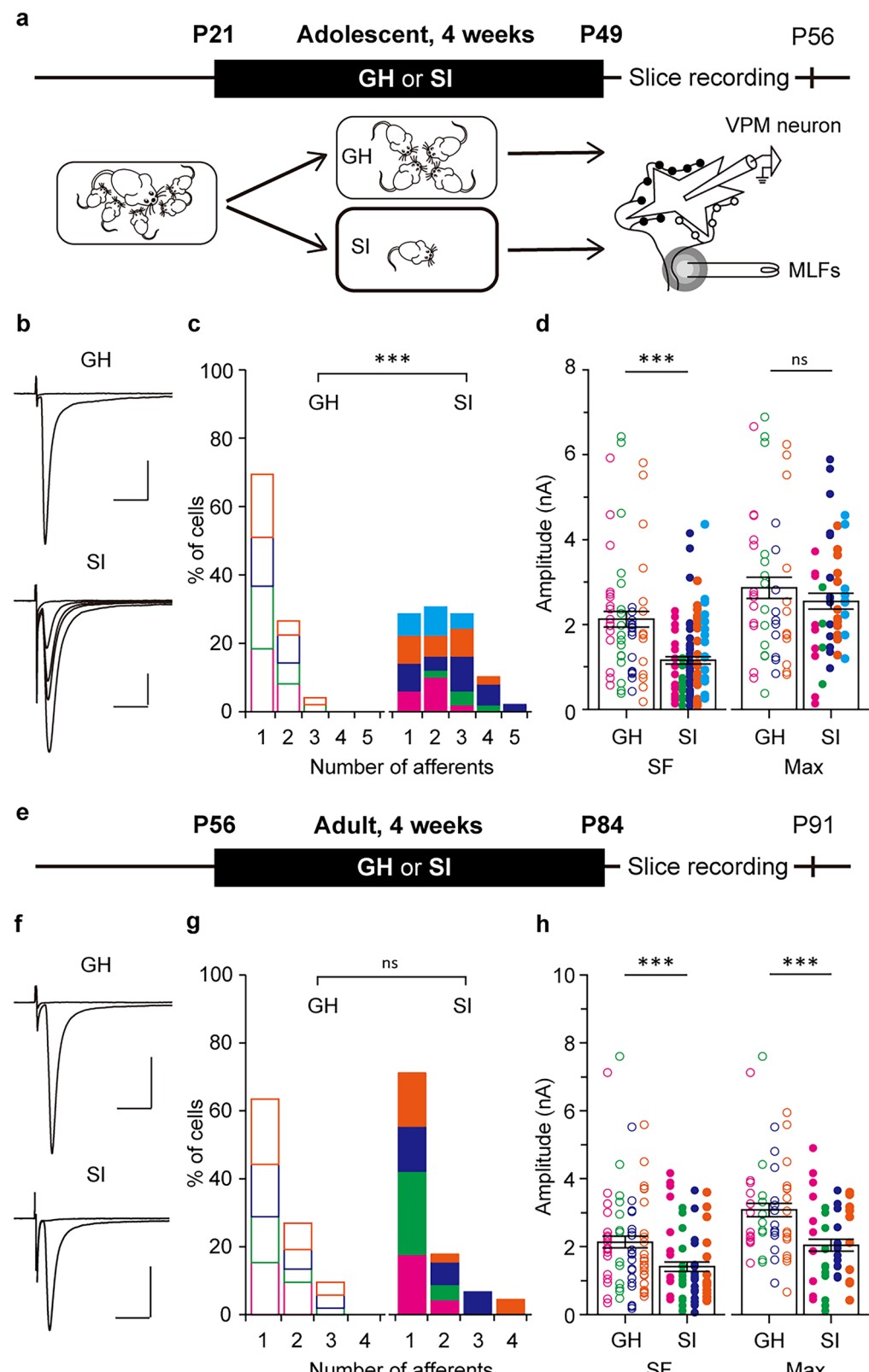

N = 15; $p$ = 0.162 by Student's $t$-test; Adult-GH, 35.2 ± 1.80 m, N = 15; SI, 37.9 ± 2.18 m, N = 14; $p$ = 0.343 by Student's $t$-test; Fig. 2c; Center time: Adolescence-GH, 55.7 ± 4.5 s, N = 16; SI, 67.1 ± 5.6 s, N = 15; $p$ = 0.121 by Student's $t$-test; Adult-GH, 85.6 ± 5.7 s, N = 15; SI, 90.5 ± 5.5 s, N = 14; $p$ = 0.541 by Student's $t$-test; Fig. 2d). In the tNORT, mice freely explored an arena containing two identical textured objects (G400 and G400') during the

learning session, followed by exposure to a G400 and a G240 in test 1 and a G400 and a G60 in test 2 (Fig. 2a; tNORT on day 3). The discrimination indexes of test 1 and test 2 were significantly greater than those of the learning session in adolescent GH mice (Learning, −0.04 ± 0.07; Test 1, 0.32 ± 0.08; Test 2, 0.34 ± 0.08; N = 16; between sessions: F(2, 30) = 9.341, $p$ < 0.001 by one-way RM-ANOVA; Learning vs. Test 1: $p$ = 0.002; Learning

**Fig. 1 | Adolescent SI rearing but not adulthood SI remodels MLF-VPM synapses.**
**a** Schematic diagram of the adolescent GH or SI rearing experiment. Male mice were weaned at P21 and divided into two rearing conditions, GH and SI. After 4 weeks in each rearing condition, electrophysiological experiments were conducted (GH, N = 4; SI, N = 5). **b** Representative traces of MLF-VPM EPSCs. **c** Histograms of afferent numbers in each condition (GH, n = 49 cells; SI, n = 49 cells; ***p < 0.001 by U-test) **d** Plots of SF- and Max-amplitudes (SF-amplitude: GH, n = 66 fibers; SI, n = 108 fibers; ***p < 0.001 by U-test; Max-amplitude: GH, n = 49 cells; SI, n = 49 cells; p = 0.637 by U-test). **e** Schematic diagram of the adulthood GH or SI rearing

experiment. Mice were weaned at P21, reared in the GH condition until P56 (8 weeks old), and then divided into GH and SI (GH, N = 4; SI, N = 4). **f** Representative records of MLF-VPM EPSCs. **g** Histograms of afferent numbers (GH, n = 52; SI, n = 45; p = 0.544 by U-test). **h** Plots of SF- and Max-amplitude (SF-amplitude: GH, n = 65; SI, n = 75; ***p < 0.001 by U-test; Max-amplitude: GH, n = 52; SI, n = 45; ***p < 0.001 by U-test). Scale bars: 5 ms, 1 nA. Holding potential: Vh = −70 mV. Each point in **d**, **h** represents the amplitude of an individual fiber and VPM neuron, respectively. Data from different mice are shown using distinct symbols (**c**, **d**, **g**, **h**). Boxes indicate mean values with error bars showing ± SEM (**d**, **h**).

vs. Test 2: p = 0.002; Test 1 vs. Test 2, p = 0.876 by Holm-Sidak post-hoc test; Fig. 2e). In adolescent SI mice, discrimination index in test 1 did not differ from that in the learning session, but test 2 values were significantly higher (Learning, −0.07 ± 0.07; Test 1, −0.04 ± 0.08; Test 2, 0.24 ± 0.08; N = 14; F(2, 26) = 4.816, p = 0.017 by one-way RM-ANOVA; Learning vs. Test 1, p = 0.759; Learning vs. Test 2, p = 0.026; Test 1 vs. Test 2, p = 0.036 by Holm-Sidak post-hoc test; Fig. 2e). In adult GH and SI mice, discrimination indexes of both test 1 and test 2 were significantly greater than those of the learning session (GH: Learning, 0.04 ± 0.07; Test 1, 0.40 ± 0.07; Test 2, 0.37 ± 0.08; N = 13; F(2, 24) = 8.653, p = 0.001 by one-way RM-ANOVA; Learning vs. Test 1, p = 0.003; Learning vs. Test 2, p = 0.004; Test 1 vs. Test 2, p = 0.753 by Holm-Sidak post-hoc test; SI: Learning, −0.01 ± 0.09; Test 1, 0.40 ± 0.07; Test 2, 0.36 ± 0.07; N = 13; F(2, 24) = 10.653, p < 0.001 by one-way RM-ANOVA; Learning vs. Test 1, p = 0.001; Learning vs. Test 2, p = 0.002; Test 1 vs. Test 2, p = 0.721 by Holm-Sidak post-hoc test; Fig. 2g). These results indicate that mice in conditions other than adolescent SI can discriminate both fine (G400 vs. G240) and coarse (G400 vs. G60) texture differences, whereas adolescent SI mice can discriminate only the coarse difference. Following the tNORT, short-term memory was assessed using the NORT with two objects (Cube and Ball) differing in color, shape, and texture (Fig. 2a; NORT on day 4). The discrimination index of test 3 was significantly greater than that in the learning session under all conditions, indicating that the short-term memory was normal (Adolescence-GH: Learning, 0.04 ± 0.05; Test 3, 0.42 ± 0.06, N = 16, p = 0.002 by paired t-test; SI: Learning, −0.02 ± 0.05; Test 3, 0.33 ± 0.09, N = 14, p = 0.006 by paired t-test; Fig. 2f. Adult-GH: Learning, 0.01 ± 0.08; Test 3, 0.57 ± 0.06, N = 13, p < 0.001 by paired t-test; SI: Learning, 0.02 ± 0.06; Test 3, 0.47 ± 0.07, N = 13, p < 0.001 by paired t-test; Fig. 2h). These results suggest that tactile discrimination via whiskers is impaired by adolescent, but not adult, SI rearing.

In the gap-cross test, mice were placed on a start platform facing another platform across a variable gap, and the maximum distance that a mouse with or without whiskers crossed was measured (Supplementary Fig. 6c, f). Because body weight differences between adolescent SI and GH mice may influence performance (Adolescence: SI, 23.0 ± 0.47 g, N = 14; GH, 21.5 ± 0.31 g, N = 15; p = 0.003 by U-test; Adult: SI, 25.2 ± 0.42 g, N = 14; GH, 25.8 ± 0.47 g, N = 13; p = 0.324 by U-test; Supplementary Fig. 6a, d), comparisons were made within the same mice before and after whisker trimming (Supplementary Fig. 6c, f). Adolescent GH mice could cross significantly larger gaps with whiskers than without (whisker intact, 6.17 ± 0.21 cm; no whiskers, 5.00 ± 0.25 cm; N = 15; p < 0.001 by paired t-test; Supplementary Fig. 6c), whereas adolescent SI mice showed no difference (whisker intact, 5.29 ± 0.14 cm; no whiskers, 5.11 ± 0.20 cm; N = 14; p = 0.397 by paired t-test; Supplementary Fig. 6c). Both adult GH and SI mice crossed significantly larger gap distance with whiskers than without (adult GH: whisker intact, 6.81 ± 0.17 cm; no whiskers, 5.92 ± 0.23 cm; N = 13; p = 0.002 by paired t-test; adult SI: whisker intact, 6.75 ± 0.19 cm; no whiskers, 5.50 ± 0.23 cm; N = 14; p < 0.001 by paired t-test; Supplementary Fig. 6f). These data suggest that adolescent SI mice exhibit deficits in distance estimation using whiskers.

Together, these behavioral experiments demonstrate that adolescent SI alters whisker-related behaviors, and that the restriction of these changes to adolescence corresponds to the finding that only adolescent SI, but not adult, induces remodeling of MLF-VPM synapses.

### Whisker deprivation does not induce synapse formation from new MLFs

We then investigated which aspects of adolescent SI contribute to MLF-VPM synaptic remodeling. First, we examined whether reducing whisker-mediated sensory experience under SI rearing could itself induce synaptic remodeling. Bilateral whiskers were trimmed under isoflurane anesthesia every 3 days during a 4-week adolescent GH rearing period (GH without whiskers, Fig. 3a). Control mice were treated with the same anesthesia only (GH with whiskers, Fig. 3a). In GH mice without whiskers, the SF-amplitude was slightly, but not significantly, smaller, and the Max-amplitude was significantly smaller than in GH mice with whiskers (SF-amplitude: without whiskers, 1298 ± 117 pA, n = 68, N = 4; with whiskers, 1700 ± 148 pA, n = 75, N = 4; p = 0.111 by U-test; Max-amplitude: without whiskers, 2053 ± 173 pA, n = 43, N = 4; with whiskers, 2899 ± 151 pA, n = 44, N = 4; p < 0.001 by U-test; Fig. 3b, d). However, the number of afferent fibers did not change by whisker removal (without whiskers, 1.58 ± 0.12 fibers, n = 43, N = 4; with whiskers, 1.73 ± 0.16 fibers, n = 44, N = 4; p = 0.844 by U-test; Fig. 3b, c), showing that adolescent whisker removal tended to attenuate synaptic strength but did not induce synapse formation from new MLFs. These results suggest that synapse formation by additional MLFs to VPM neurons under adolescent SI cannot be explained solely by reduced whisker-mediated sensory input.

### Transient overactivation of glucocorticoid receptors by adolescent social isolation leads to remodeling at MLF-VPM synapses

We next examined the role of stress hormones in this process. To collect samples repeatedly and noninvasively from the same mice, we measured urinary corticosterone every 7 days throughout the 4-week GH and SI periods (Fig. 4a, b). Our measurements revealed that urinary corticosterone levels in SI mice were significantly elevated compared with GH mice on day 7 after the onset of GH or SI during adolescence (adolescent GH, 84.8 ± 14.8 ng/ml, N = 12; adolescent SI, 190.4 ± 30.0 ng/ml, N = 14; p < 0.001 by U-test; Fig. 4a). This difference disappeared during the following week, and no significant differences were found between GH and SI mice after day 14 (day 14, p = 0.217; day 21, p = 0.198; day 28, p = 0.797 by U-test; GH, N = 12; SI, N = 14; Fig. 4a). In adults, no differences in corticosterone levels were detected at any time point, including day 7 (day 7, p = 0.959; day 14, p = 0.878; day 21, p = 0.083; day 28, p = 0.279 by U-test; GH, N = 8; SI, N = 8; Fig. 4b). We hypothesized that the transient corticosterone increase during the first week of SI might underlie adolescence-specific synaptic remodeling. Slice recordings revealed that synaptic remodeling had already occurred one week after the onset of adolescent SI (afferent numbers: 1w-GH, 1.32 ± 0.08 fibers, n = 44, N = 4; 1w-SI, 2.17 ± 0.17 fibers, n = 46, N = 3; p < 0.001 by U-test; SF-amplitude: 1w-GH, 1958 ± 169 pA, n = 58, N = 4; 1w-SI, 1103 ± 111 pA, n = 99, N = 3; p < 0.001 by U test; Max-amplitude: 1w-GH, 2581 ± 186 pA, n = 44, N = 4; 1w-SI, 2347 ± 160 pA, n = 46, N = 3; p = 0.475 by U-test; Fig. 4c–f). To examine the causal relationship between transient corticosterone elevation and remodeling of MLF-VPM synapses, glucocorticoid receptors (GRs) were pharmacologically activated or blocked during one week of GH and SI rearing, and electrophysiological experiments were conducted the following week (Fig. 5a–h). In GH mice intraperitoneally (ip) administered with the GR agonist dexamethasone (DEX; 5 mg/kg body weight (bw), once a day), afferent number significantly increased, SF-amplitude significantly

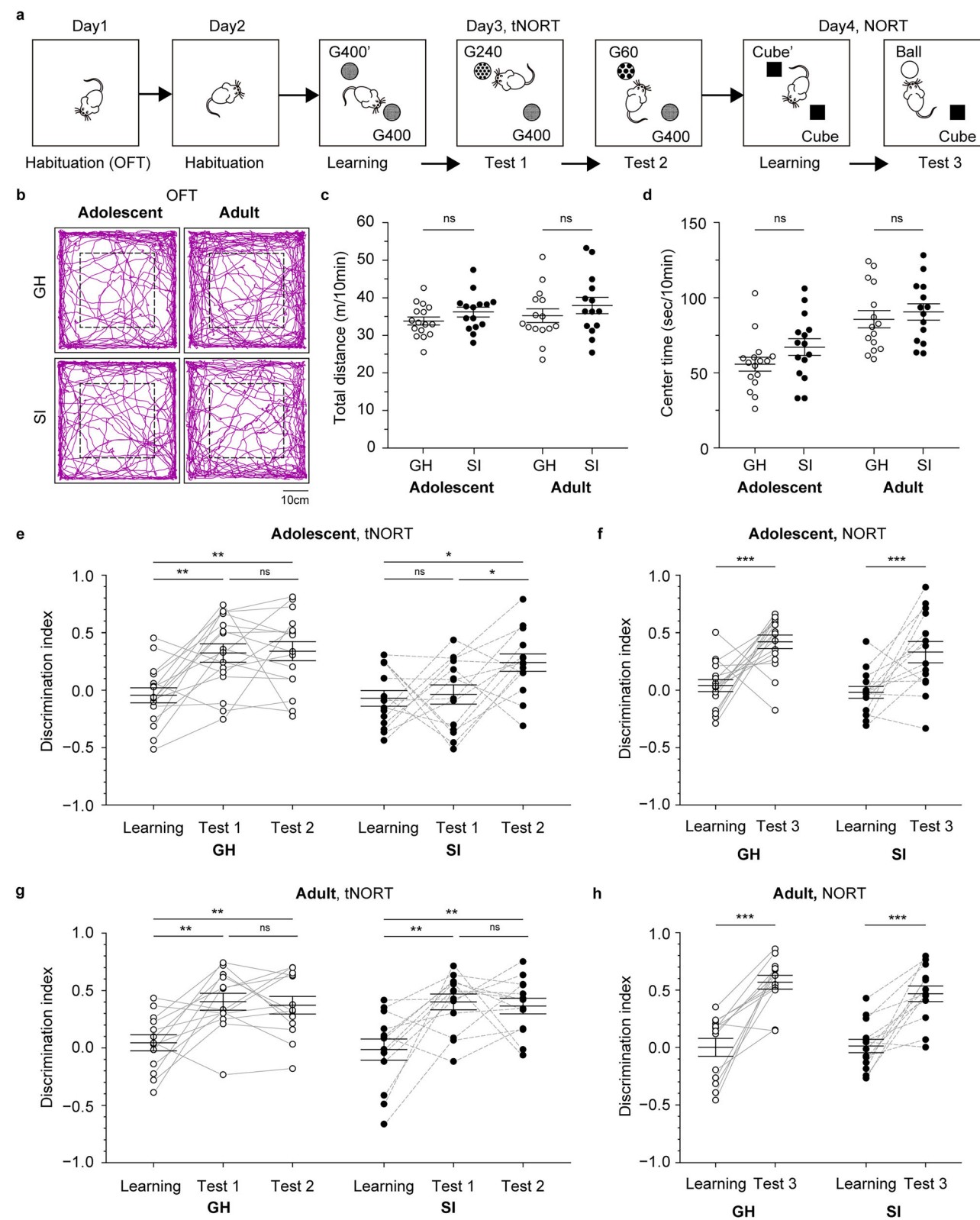

decreased, and Max-amplitude remained unchanged (afferent number: GH + VHCL, 1.38 ± 0.11 fibers, n = 39, N = 3; GH + DEX, 2.5 ± 0.21 fibers, n = 40, N = 4; $p < 0.001$ by $U$-test; SF-amplitude: GH + VHCL, 2221 ± 210 pA, n = 54, N = 3; GH + DEX, 1243 ± 99 pA, n = 95, N = 4; $p < 0.001$ by $U$-test; Max-amplitude: GH + VHCL, 3075 ± 267 pA, n = 39, N = 3; GH + DEX, 3046 ± 219 pA, n = 40, N = 4; $p = 0.977$ by $U$-test; Fig. 5a–d). Conversely, the afferent number was significantly reduced in SI mice treated

with the GR antagonist mifepristone (RU486; 40 mg/kg bw, once a day) compared with VHCL-treated SI mice (SI + RU486, 1.42 ± 0.10 fibers, n = 48, N = 4; SI + VHCL, 2.22 ± 0.18 fibers, n = 45, N = 3; $p < 0.001$ by $U$-test; Fig. 5e–g). SF-amplitude was moderately larger but not significantly different, while Max-amplitude remained lower in SI + RU486 than in SI-VHCL mice (SF-amplitude: SI + RU486, 1820 ± 190 pA, n = 68, N = 4; SI + VHCL, 1495 ± 109 pA, n = 100, N = 3; $p = 0.479$ by $U$-test; Max-

**Fig. 2 | Social isolation during adolescence but not adulthood disrupts texture discrimination. a** Experimental flow. Male mice were used. **b** Trajectories of mice in the area during OFT. The area surrounded by a dashed line was where we measured the center time in (**c**). **c** Total distance mice traveled during the OFT (Adolescent: GH, N = 16; SI, N = 15; df = 29; t = −1.436; p = 0.162 by Student's t-test; Adult: GH, N = 15; SI, N = 14; df = 27; t = −0.964; p = 0.343 by Student's t-test). **d** Time spent in the center area during the OFT (Adolescent: GH, N = 16; SI, N = 15; df = 29; t = −1.596; p = 0.121 by Student's t-test; Adult: GH, N = 15; SI, N = 14; df = 27; t = −0.619; p = 0.541 by Student's t-test). **e** Discrimination index in the tNORT of adolescent GH or SI condition (GH, N = 16; between sessions: F(2, 30) = 9.341, p < 0.001 by one-way RM-ANOVA; Learning vs. Test 1: **p = 0.002; Learning vs. Test 2: **p = 0.002; Test 1 vs. Test 2, p = 0.876 by Holm-Sidak post-hoc test; SI, N = 14; between sessions: F(2, 26) = 4.816, *p = 0.017 by one-way RM-ANOVA; Learning vs. Test 1: p = 0.759; Learning vs. Test 2: *p = 0.026, Test 1 vs. Test 2: *p = 0.036 by Holm-Sidak post-hoc test. **f** Discrimination index in the NORT of adolescent GH or SI condition (GH, N = 16; df = 15; t = −3.745; **p = 0.002 by paired t-test; SI, N = 14; df = 13; t = −3.283; **p = 0.006 by paired t-test). **g** Discrimination index in the tNORT of adult GH or SI condition (GH, N = 13; between sessions: F(2, 24) = 8.653, p = 0.001 by one-way RM-ANOVA; Learning vs. Test 1: **p = 0.003; Learning vs. Test 2: **p = 0.004; Test 1 vs. Test 2: p = 0.753 by Holm-Sidak post-hoc test; SI, N = 13; between sessions: F(2, 24) = 10.653, ***p < 0.001 by one-way RM-ANOVA; Learning vs. Test1: **p = 0.001; Learning vs. Test 2: **p = 0.002; Test 1 vs. Test 2: p = 0.721 by Holm-Sidak post-hoc test). **h** Discrimination index in the NORT of adult GH and SI mice (GH, N = 13; df = 12; t = −5.883; ***p < 0.001 by paired t-test; SI, N = 13; df = 12; t = −5.405; ***p < 0.001 by paired t-test). Data from the same mouse are connected by gray solid (GH) or dashed (SI) lines. Vertical bars represent mean ± SEM (**c–h**).

amplitude: SI + RU486, 2578 ± 260 pA, n = 48, N = 4; SI + VHCL, 3322 ± 229 pA, n = 45, N = 3; p = 0.013 by U-test; Fig. 5h). Furthermore, SI mice treated with RU486 for the first week and then kept in SI for an additional three weeks without drug administration retained a significantly smaller number of afferents than VHCL-treated mice (SI + 1w-RU486, 1.20 ± 0.06 fibers, n = 41, N = 4; SI + 1w-VHCL, 2.00 ± 0.14 fibers, n = 38, N = 4; p < 0.001 by U-test; Fig. 5i–k). In these conditions, SF-amplitude was significantly greater in SI + 1w-RU486 than in SI + 1w-VHCL (SI + 1w-RU486, 1959 ± 183 pA, n = 49, N = 4; SI + 1w-VECL, 1365 ± 121 pA, n = 76, N = 4; p = 0.005 by U-test; Fig. 5l). The difference in Max-amplitude observed immediately after 1w-RU486 administration (Fig. 5h) was no longer present (SI + 1w-RU486, 2341 ± 215 pA, n = 41, N = 4; SI + 1w-VHCL, 2729 ± 211 pA, n = 38, N = 4; p = 0.209 by U-test; Fig. 5l). Collectively, these results suggest that corticosterone levels are transiently elevated during the first week of adolescent SI, and that this elevation triggers MLF-VPM synaptic remodeling through a GR-mediated mechanism.

To further test the relationship between elevated corticosterone levels and synaptic reorganization, we examined the effects of DEX in adult GH mice (Supplementary Fig. 7). Adult male mice were treated with DEX once a day for 1 week to activate GRs. DEX-treated adult mice showed a significant increase in afferent number (GH + VHCL, 1.52 ± 0.13 fibers, n = 50, N = 4; GH + DEX, 2.09 ± 0.15 fibers, n = 45, N = 4; p = 0.001 by U-test; Supplementary Fig. 7a–c). SF-amplitude was significantly reduced, but Max-amplitude remained unchanged in GH + DEX mice (SF-amplitude: GH + VHCL, 1826 ± 142 pA, n = 76, N = 4; GH + DEX, 1153 ± 93 pA, n = 94, N = 4; p < 0.001 by U-test; Max-amplitude: GH + VHCL, 2775 ± 155 pA, n = 50, N = 4; GH + DEX, 2409 ± 184 pA, n = 45, N = 4; p = 0.084 by U-test; Supplementary Fig. 7d). These results support the idea that elevated corticosterone levels can induce remodeling at MLF-VPM synapses, and further suggest that these synapses maintain plasticity for remodeling in adulthood when GRs are strongly activated.

### GRs expressed in VPM neurons are responsible for the remodeling of MLF-VPM synapses induced by adolescent SI

We aimed to identify and localize the GRs responsible for synaptic remodeling during adolescent SI. In wild-type mice at P21, strong GR immunolabeling was observed in the hippocampal CA1 and in several other brain regions, including the thalamic VPM (Supplementary Fig. 8a, b). We counted neurons, astrocytes, and microglia that were triple-positive for a cell-type marker, DAPI, and GR, because 96% of GR-labeled structures were co-labeled with DAPI in the VPM (96.0 ± 0.44%, N = 3; Supplementary Fig. 8c, d). Quantitative analysis revealed that 98% of NeuN-positive neurons, 83% of S100B-positive astrocytes, and 79% of IBA1-positive microglia expressed GRs (neuron, N = 3; astrocyte, N = 3; microglia, N = 3; Supplementary Fig. 8e–n). Among these cell types, we generated thalamic neuron-specific GR knockout (GR-ThKO) mice by crossing floxed-GR mice[35] with 5HTT-Cre mice[36,37], and examined the role of neuronal GRs in the VPM in synaptic remodeling and behavioral changes under adolescent SI (Fig. 6). Floxed-GR mice exhibited GR immunolabeling comparable to

that of wild-type mice (Fig. 6a; Supplementary Fig. 8a). In contrast, GR immunoreactivity in the VPM of GR-ThKO mice was markedly reduced but not completely abolished (Fig. 6a, b). In the VPM of GR-ThKO mice, GR-expressing neurons almost entirely absent, while GR-expressing astrocytes and microglia remained unchanged, confirming that GR deletion occurred selectively in neurons (neuron: floxed-GR, 97.7 ± 0.52%, N = 3; GR-ThKO, 3.22 ± 0.33%, N = 4; p < 0.001 by Student's t-test; astrocyte: floxed-GR, 87.9 ± 2.82%, N = 3; GR-ThKO, 84.2 ± 1.32%, N = 3, p = 0.300 by Student's t-test; microglia: floxed-GR, 81.5 ± 3.77%, N = 3; GR-ThKO, 81.0 ± 1.79%, N = 3; p = 0.146 by Student's t-test; Fig. 6c–i).

Male floxed-GR mice and GR-ThKO mice were reared in SI conditions for 4 weeks starting at P21 (Fig. 6j–m). GR-ThKO mice exhibited significantly fewer afferent fibers than floxed-GR mice (GR-ThKO, 1.62 ± 0.12 fibers, n = 47, N = 4; floxed-GR, 2.10 ± 0.11 fibers, n = 56, N = 4; p = 0.005 by U-test; Fig. 6k, l). There were no significant differences in the SF- or the Max-amplitudes between genotypes (SF-amplitude: GR-ThKO, 1412 ± 130 pA, n = 76, N = 4; floxed-GR, 1254 ± 101 pA, n = 117, N = 4; p = 0.246 by U-test; Max-amplitude: GR-ThKO, 2284 ± 201 pA; floxed-GR, 2620 ± 180 pA, n = 56, N = 4; p = 0.146 by U-test; Fig. 6m). These electrophysiological results indicate that synapse formation of additional MLF afferents under adolescent SI requires activation of GRs in VPM neurons, while synaptic strength at individual MLF-VPM connections is not sololey controlled by neuronal GR signaling.

We next compared whisker-related behavioral performance between floxed-GR and GR-ThKO male mice experienced adolescent SI (Supplementary Fig. 9). Total distance traveld and time spent in the center area during the open-field test did not differ between genotypes (total distance: floxed-GR, 46.8 ± 2.52 m, N = 9; GR-ThKO, 52.1 ± 3.96 m, N = 10; p = 0.285 by Student's t-test; center time: floxed-GR, 72.9 ± 5.65 s, N = 9; GR-ThKO, 89.9 ± 9.20 s, N = 10; p = 0.143 by Student's t-test; Supplementary Fig. 9a–c), suggesting that locomotor activity and anxiety-like behavior were not alterd in GR-ThKO mice. In the tNORT, floxed-GR mice showed significantly higher discrimination indexes only in test 2, but not test 1, compared with the learning session, as also observed in wild-type mice experienced adolescent SI shown in Fig. 2e (learning, −0.05 ± 0.09; test 1, 0.12 ± 0.10; test 2, 0.31 ± 0.11; N = 9; F(2, 16) = 6.519, p = 0.008 by one-way RM-ANOVA; learning vs. test 1, p = 0.102; learning vs. test 2, p = 0.007; test 1 vs. test 2, p = 0.151 by Holm-Sidak post-hoc test; Supplementary Fig. 9d). In contrast, GR-ThKO mice showed significantly larger discrimination indexes in both test 1 and test 2 than in the learning session (learning, −0.09 ± 0.08; test 1, 0.28 ± 0.10; test 2, 0.38 ± 0.07; N = 10; F(2, 18) = 7.075, p = 0.005 by one-way RM-ANOVA; learning vs. test 1, p = 0.023; learning vs. test 2, p = 0.007; test 1 vs. test 2, p = 0.459 by Holm-Sidak post-hoc test; Supplementary Fig. 9d), indicating that the deterioration of texture discrimination caused by adolescent SI was prevented in GR-ThKO mice. In the NORT, both genotypes showed normal short-term memory (floxed-GR: learning, 0.01 ± 0.11; test 3, 0.36 ± 0.12; N = 9, p = 0.004 by paired t-test; GR-ThKO: learning, -0.05 ± 0.07; test 3, 0.40 ± 0.08; N = 10, p < 0.001 by paired t-test; Supplementary Fig. 9e). Similarly, gap-cross performance was significantly improved in GR-ThKO mice (GR-ThKO, 6.63 ± 0.26 cm,

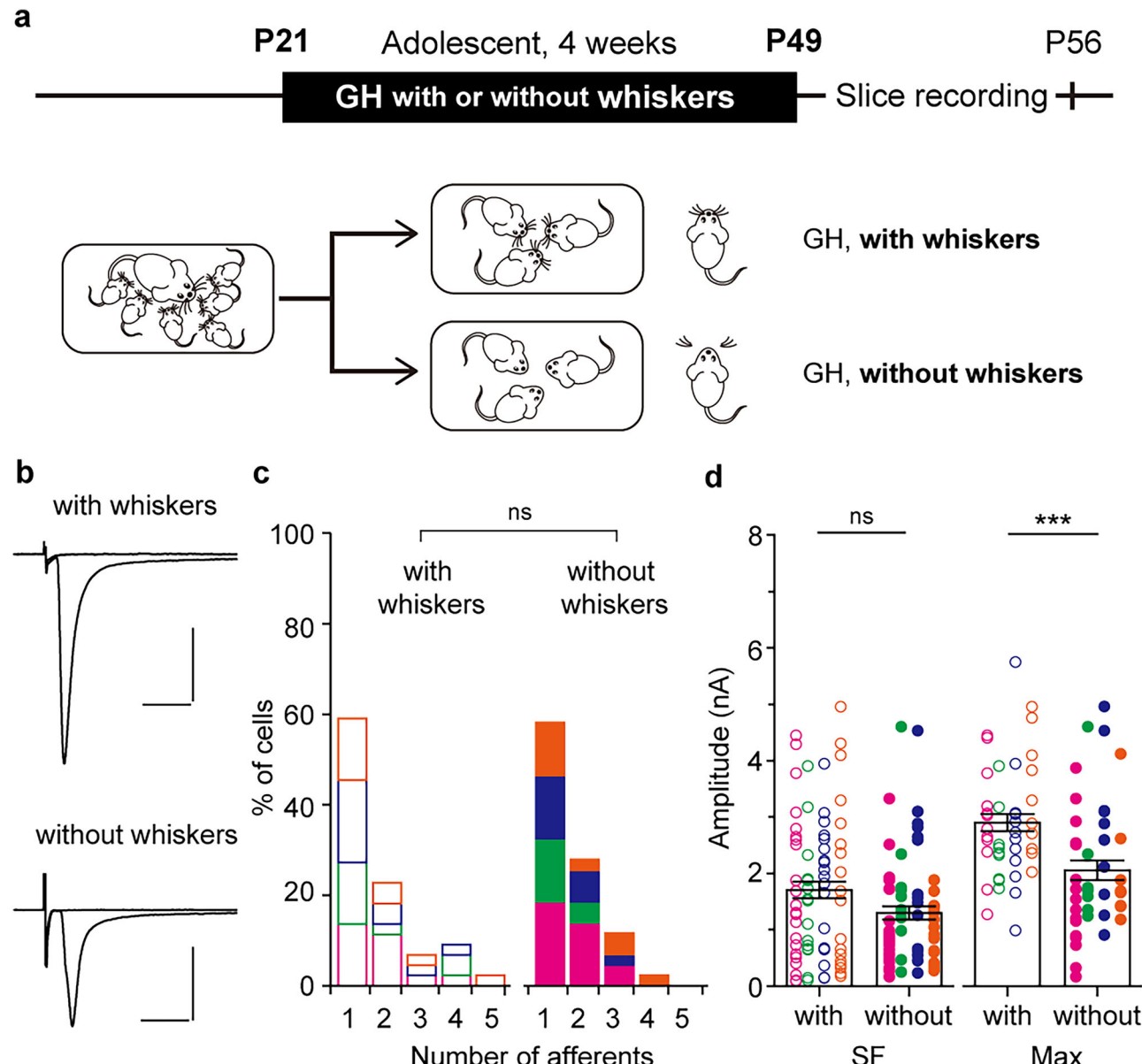

**Fig. 3 | Whisker deprivation itself does not increase afferent numbers to VPM neurons. a** Mice were weaned at P21 and divided into two conditions: GH with whiskers and GH without whiskers. Mice in the GH without whiskers had their whiskers trimmed under isoflurane anesthesia every 3 days for 4 weeks, while mice in the GH with whiskers underwent anesthesia only (GH with whiskers, N = 4; GH without whiskers, N = 4). **b** Representative MLF-VPM EPSCs traces in each condition. **c** Histograms of afferent numbers (GH with whiskers, n = 44; GH without whiskers, n = 43; p = 0.844 by U-test). **d** Plots of SF- and Max-amplitudes (SF-amplitude: GH with whiskers, n = 75; GH without whiskers, n = 68; p = 0.111 by U-test; Max-amplitude: GH with whiskers, n = 44; GH without whiskers, n = 43; ***p < 0.001 by U-test). Scale bars: 5 ms, 1 nA. Vh = −70 mV. Each point in **d** represents the amplitude of an individual fiber and VPM neuron, respectively. Boxes indicate mean values with error bars showing ± SEM (**d**). Data from different mice are shown using distinct symbols (**c, d**). All data were obtained from male mice.

N = 15; floxed-GR, 5.56 ± 0.14 cm, N = 18; p = 0.002 by U-test; Supplementary Fig. 9g). These behavioral data suggest that GRs in thalamic neurons contribute to the modulation of whisker-related behaviors following adolescent SI. While we cannot exclude the possibility that GR deletion also occurred in brain regions outside the VPM[36], and that such changes could directly or indirectly influence performance in the tNORT and gap-crossing tests, the present data tentatively support the notion that GRs in VPM neurons play a role in shaping whisker-related behavioral outcomes following adolescent SI. Regarding tactile thresholds, von Frey test results in GR-ThKO mice were not different from that in floxed-GR mice (Supplementary Fig. 9h). These findings suggest that alterations in tactile threshold observed in this study are not solely attributable to GR-dependent remodeling of MLF–VPM synapses, but may also involve additional mechanisms induced by adolescent SI.

## Cohabitation with a male during adolescence partially normalizes the remodeled synapses

Finally, we tested whether the remodeling of MLF-VPM synapses induced by adolescent SI could be normalized by the rearing environment. Male mice were cohabited with another mouse (CH) for 4 weeks following either 4 weeks or 1 week of adolescent SI (4w- or 1w-SI + 4w-CH, Fig. 7a). In the present study, age-matched male GH mice were used as cohabitants for SI mice, and age-matched SI mice (4w- or 1w-SI + 4w-SI) were used as controls. The number of afferents was not significantly different between 4w-SI + 4w-CH and 4w-SI + 4w-SI mice (4w-SI + 4w-CH, 1.86 ± 0.16, n = 43, N = 4; 4w-SI + 4w-SI, 2.14 ± 0.15, n = 37, N = 3; p = 0.103 by U-test; Fig. 7b, c). SF- and Max-amplitudes also did not differ significantly (SF-amplitude: 4w-SI + 4w-CH, 1305 ± 137 pA, n = 80, N = 4; 4w-SI + 4w-SI, 1254 ± 112 pA, n = 78, N = 3; p = 0.776 by U-test; Max-amplitude: 4w-

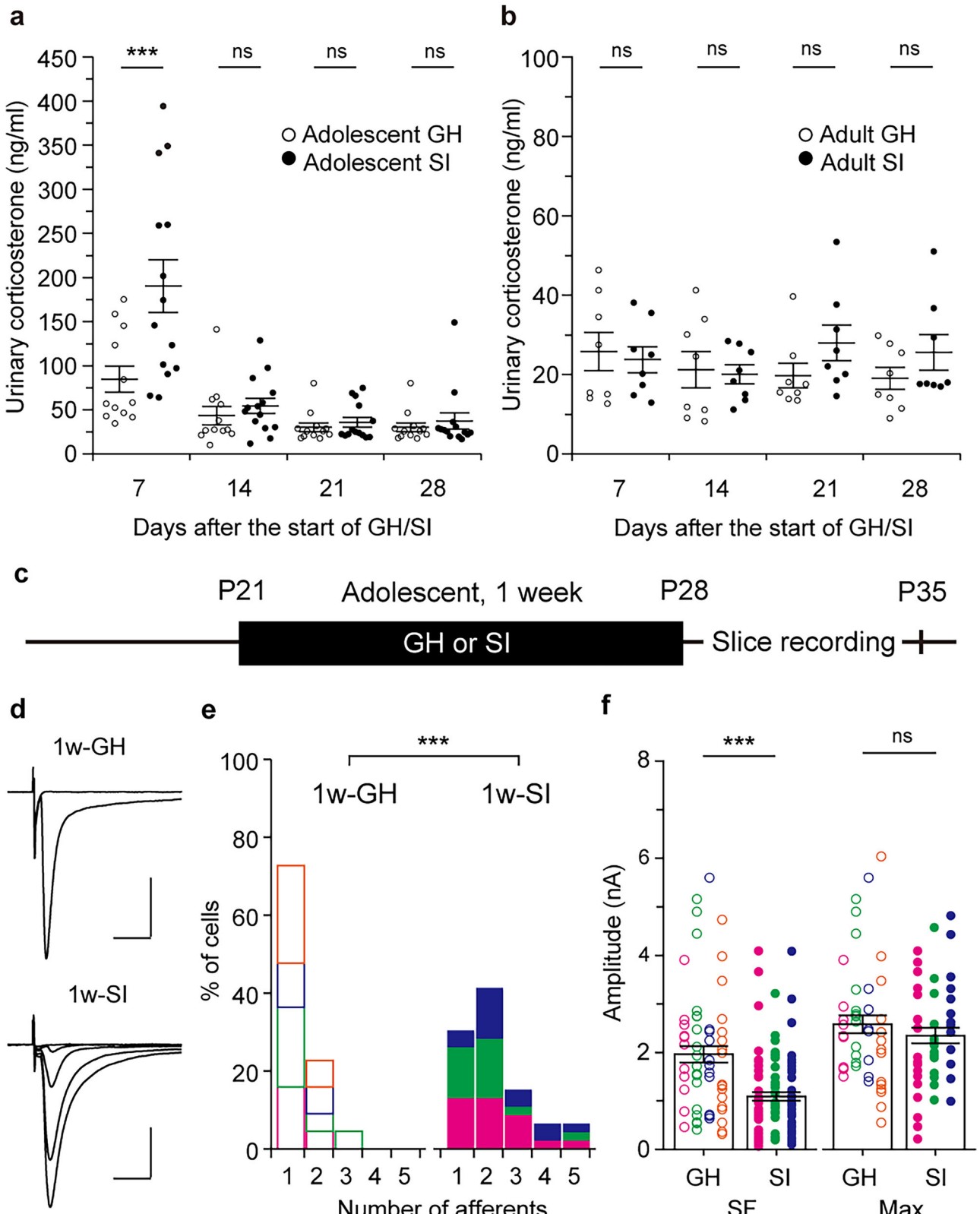

**Fig. 4 | Adolescent SI rearing causes a transient elevation of corticosterone, and synapse remodeling occurs within the first week of the SI period. a, b** Urinary corticosterone levels measured during GH or SI rearing starting at P21 (**a**, Adolescent GH, N = 12; Adolescent SI, N = 14; day 7, ***p < 0.001; day 14, p = 0.21; day 21, p = 0.198; day 28, p = 0.797 by U-test) and 8 weeks old (**b**, Adult GH, N = 8; Adult SI, N = 8; day 7, p = 0.959; day 14, p = 0.878; day 21, p = 0.083; day 28, p = 0.279 by U-test). Solid lines represent mean ± SEM. **c** Mice were weaned at P21 and reared under GH or SI conditions for 1 week (1w-GH, N = 4; 1w-SI, N = 3). **d** Representative MLF-VPM

EPSC traces. **e** Histograms of afferent numbers (1w-GH, n = 44; 1w-SI, n = 46; ***p < 0.001 by U-test). **d** Plots of SF- and Max-amplitude (SF-amplitude: 1w-GH, n = 58; 1w-SI, n = 99; ***p < 0.001 by U-test; Max-amplitude: 1w-GH, n = 44; 1w-SI, n = 46; p = 0.475 by U-test). Scale bars: 5 ms, 1 nA. Vh = −70 mV. Each point in **f** represents the amplitude of an individual fiber and VPM neuron, respectively. Boxes indicate mean values with error bars showing ± SEM (**f**). Data from different mice were shown using distint symbols (**e, f**). All data were obtained from male mice.

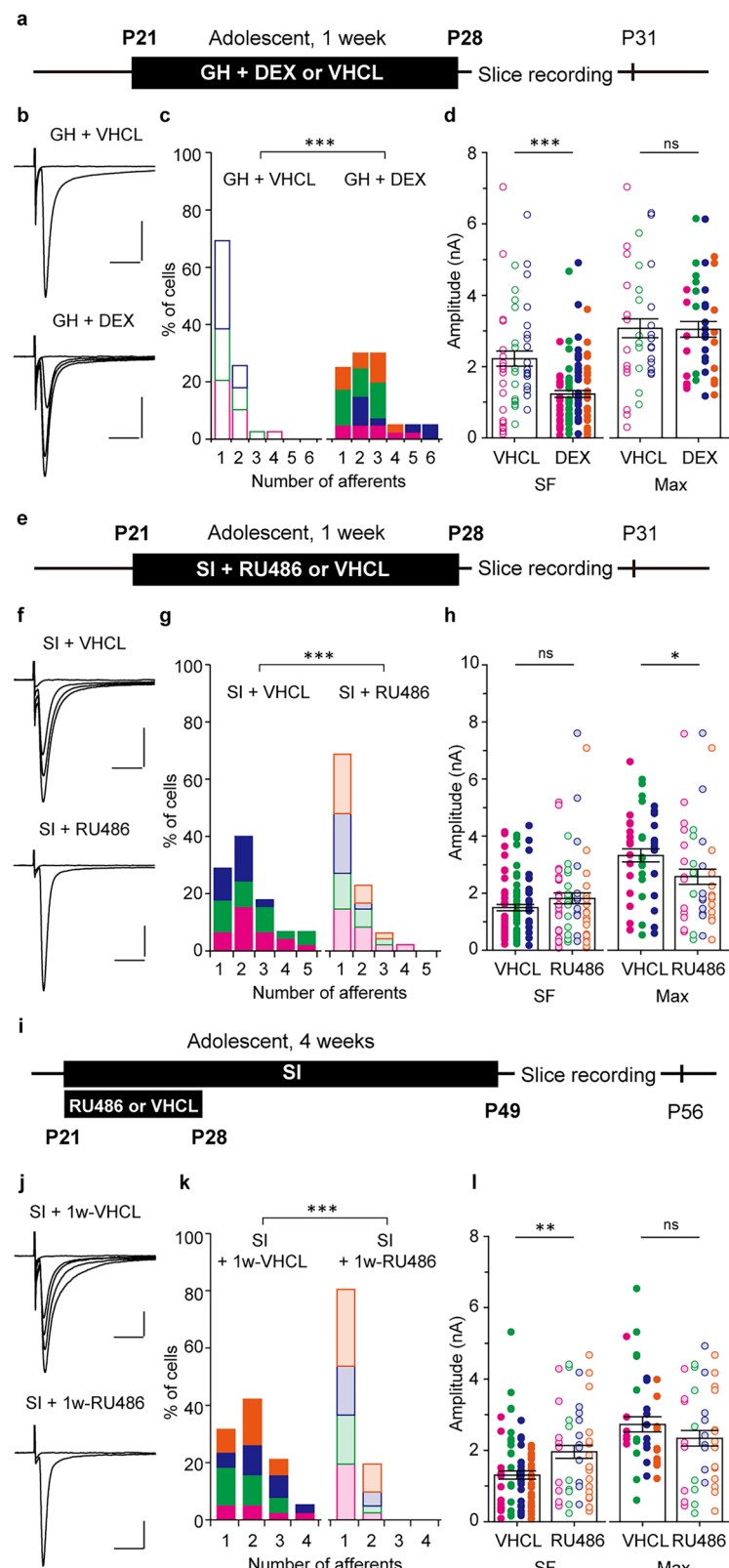

SI + 4w-CH, 2429 ± 249.9 pA, n = 43, N = 4; 4w-SI + 4w-SI, 2643 ± 227 pA, n = 37, N = 3; $p$ = 0.300; Fig. 7d). By contrast, when 4w-CH started after only 1 week of SI (Fig. 7e), the number of afferent fibers decreased significantly, and SF-amplitude increased in 1w-SI + 4w-CH mice compared to 1w-SI + 4w-SI mice (afferent number: 1w-SI + 4w-CH, 1.97 ± 0.17, n = 38, N = 4; 1w-SI + 4w-SI, 2.52 ± 0.19, n = 44, N = 4; $p$ = 0.042 by $U$-test;

Fig. 7f, g; SF-amplitude: 1w-SI + 4w-CH, 1471 ± 290.3 pA, n = 75, N = 4; 1w-SI + 4w-SI, 1218 ± 104.8 pA, n = 111, N = 4; $p$ = 0.018 by $U$-test; Max-amplitude: 1w-SI + 4w-CH, 2903 ± 190.6 pA, n = 38, N = 4; 1w-SI + 4w-SI, 3072 ± 260.5 pA, n = 44, N = 4; $p$ = 0.838 by $U$-test; Fig. 7h). These results suggest that the remodeled synapses can be partially normalized if SI mice are cohabited with other mice during adolescence. However,

**Fig. 5 | Pharmacological manipulation of GRs during the first week of adolescent isolation can induce or prevent remodeling of MLF-VPM synapses. a** Mice were weaned at P21 and kept under GH conditions for 1 week. During this period, DEX (5 mg/kg bw) or an equal volume of VHCL was administered intraperitoneally once a day (GH + VHCL, N = 3; GH + DEX, N = 4). **b** Representative MLF-VPM EPSCs. **c** Histograms of afferent numbers in each condition (GH + VHCL, n = 39; GH + DEX, n = 40; ***$p < 0.001$ by $U$-test). **d** Plots of SF- and Max-amplitudes (SF-amplitude: GH + VHCL, n = 54; GH + DEX, n = 96; *** $p < 0.001$ by $U$-test; Max-amplitude: GH + VHCL, n = 39; GH + DEX, n = 40; $p = 0.977$ by $U$-test). **e** Mice were weaned at P21 and reared in SI conditions for 1 week with daily intraperitoneal injections of RU486 (40 mg/kg bw) or VHCL (SI + VHCL, N = 3; SI + RU486, N = 4). **f** Representative MLF-VPM EPSCs. **g** Histograms of afferent numbers (SI + VHCL, n = 45; SI + RU486, n = 48; ***$p < 0.001$ by $U$-test). **h** Plots of SF- and Max-amplitudes (SF-amplitude: SI + VHCL, n = 100; SI + RU486, n = 68; $p = 0.479$ by $U$-test; Max-amplitude: SI + VHCL, n = 45; SI + RU486, n = 48; *$p = 0.013$ by $U$-test). **i** Mice were weaned at P21 and housed in SI conditions for 4 weeks. RU486 (40 mg/kg bw) or VHCL was intraperitoneally injected only during the first week (SI + 1w-VHCL, N = 4; SI + 1w-RU486, N = 4). **j** Representative MLF-VPM EPSCs. **k** Histograms of afferent numbers (SI + 1w-VHCL, n = 38, SI + 1w-RU486, n = 41; ***$p < 0.001$ by $U$-test). **l** Plots of SF- and Max-amplitudes (SF-amplitude: SI + 1w-VHCL, n = 76, SI + 1w-RU486, n = 49; **$p = 0.005$ by $U$-test; Max-amplitude: SI + 1w-VHCL, n = 38, SI + 1w-RU486, n = 41; $p = 0.209$ by $U$-test). Scale bars: 5 ms, 1 nA. Vh = −70 mV. Each point in **d**, **h**, **l** represents the amplitude of an individual fiber and VPM neuron, respectively. Boxes indicate mean values with error bars showing ± SEM (**d**, **h**, **l**). Data from different mice are shown using distinct symbols (**c**, **d**, **g**, **h**, **k**, **l**). All data were obtained from male mice.

when compared to the adolescent GH mice (Fig. 1a–d), the number of afferents in the 1w-SI + 4w-CH mice was still significantly higher ($p = 0.003$ by $U$-test), and their SF-amplitude remained significantly lower ($p = 0.005$ by $U$-test), indicating that CH during adolescence is insufficient for full recovery. Identifying the factors required for complete normalization of thalamic synapses is one of an important direction for future research.

## Discussion

In the present study, we provided the first evidence that SI during adolescence largely remodels MLF-VPM synapses in the somatosensory thalamus. This remodeling occurred only when isolation was experienced during adolescence, but not during adulthood. Consistent with the synaptic remodeling, adolescent SI mice showed impairments in tactile discrimination and distance estimation using their whiskers. A series of experiments using wild-type mice administered with GR agonist and antagonist, together with GR-ThKO mice, demonstrated that synaptic remodeling is induced via a mechanism involving GRs activated by a transient elevation of corticosterone during the first week of isolation. Furthermore, cohabitation with another mouse during adolescence partially restored this remodeling. Collectively, our findings reveal adolescents' vulnerability to SI, highlighting a critical risk of disrupting neural circuits in the somatosensory thalamus through strong GR activation and impaired tactile-based behaviors.

What is the signaling cascade underlying this synaptic remodeling? Adolescent SI produced two changes at MLF-VPM synapses: an increase in the number of afferents and a decrease in SF-amplitude (Fig. 1b–d). Both changes were induced by DEX administration under GH conditions, regardless of developmental stage (Fig. 5a–d, Supplementary Fig. 7), suggesting that they commonly depend on GR activation. In contrast, adult SI did not increase the number of afferents in either sex, but only reduced SF-amplitude (Fig. 1e–h, Supplementary Fig. 4e–h). This observation suggests that GR activation underlies both processes, but with different thresholds for driving. Alternatively, since only the increase in afferents was suppressed in GR-ThKO mice (Fig. 6k–m), the increase in afferents depends on GRs expressed in neurons. At the same time, the reduction of SF-amplitude may involve GRs expressed in other cell types. Indeed, in the VPM, over 80% of astrocytes and microglia expressed GR as well as in nearly all neurons (Supplementary Fig. 8), suggesting potential contributions of glial GRs to the synaptic remodeling.

A factor controlling both the increase in afferent number and the attenuation of SF-amplitude is the enhancement of GABA$_A$R-mediated tonic inhibition of VPM neurons, as previously shown in infraorbital nerve-cut (IONC) mice[37,38]. The IONC strengthens tonic inhibition on VPM neurons by increasing the expression of α4 subunit-containing GABA$_A$Rs[28]. Similarly, adolescent SI (P30-P60) has been reported to increase tonic inhibition by upregulating the expression of GABA$_A$Rs containing α4 and δ subunits in the rat hippocampus[38,39]. In addition to altered expression level, GABA$_A$R function can be regulated by allosteric modulators. Thalamo-cortical neurons retain the ability to synthesize positive allosteric modulators of GABA$_A$Rs during adolescence[40]. In the prefrontal cortex, adolescent SI reduces allopregnanolone-mediated positive allosteric modulation of GABA$_A$Rs by downregulating its rate-limiting enzyme, a

mechanism implicated in autism spectrum disorder-like behavior[41,42]. Whether adolescent SI regulates tonic inhibition of VPM neurons, for example, through increased expression of α4/δ GABA$_A$R subunits or changes in the synthesis of allosteric modulators, and whether these changes result from GR activation, remain important questions to be investigated.

Mice experienced social isolation during adolescence showed altered performance in both the tNORT and the gap-cross tests. How might the remodeling of the MLF-VPM synapses account for behavioral changes? Considering that the tNORT and gap-cross tests require the use of whiskers to discriminate texture and estimate distance, the poor performance observed after adolescent SI may indicate that the neural circuits underlying fine object recognition using whiskers are not functioning correctly. The tactile ascending pathway from whiskers to the S1 possesses unique structural and functional features. Neurons in the brainstem PrV2, the thalamic VPM, and S1 layer 4 each respond primarily to a specific whisker (a principal whisker), and cells with a common principal whisker are clustered together in a pattern that reproduces the whisker arrangement, called barrels, barrelettes, and barreloids, respectively[17,18]. Disruption of this somatotopic organization impairs whisker-dependent perception. For instance, sensory thalamus-specific NR1-null mice, which display hypoplasia of the barreloids and barrels with reduced S1 activation following whisker stimulation, show behavioral abnormalities in both tNORT and gap-cross tests similar to adolescent SI mice[36]. Likewise, mice with reduced firing probability of VPM neurons in response to afferent input exhibit abnormal tactile discrimination[31]. These previous reports suggest that the reduced SF-amplitude observed in VPM neurons of adolescent SI mice (Fig. 1d) may diminish excitatory outputs to S1, thereby raising the threshold for recognizing features of objects with whiskers. However, in our experiments, adult SI mice also showed reduced (Fig. 1h) but perform normally in the tNORT and the gap-cross test (Fig. 2g, Supplementary Fig. 6e). Taken together, our results suggest that the decreased SF-amplitude alone is not enough to disrupt these whisker-related behaviors; rather, the combination of reduced synaptic efficacy and increased afferents during adolescent SI may be necessary to alter behaviors.

In addition to the reduction in SF-amplitude, adolescent SI also induced the formation of new afferent synapses (Fig. 1c). Synapse formation by new afferents to VPM neurons has previously been reported after IONC[23]. In IONC mice, the newly recruited afferents arise from ectopic regions outside the PrV2[23]. In contrast, our analysis showed that the origin of the newly formed synapses under adolescent SI is PrV2 rather than non-PrV2 regions (Supplementary Fig. 5). Assuming that the newly added afferents are axons of PrV2 neurons corresponding to different principal whiskers, the tuning of S1 neurons to their principal whiskers may become broadened. If this occurs, the accuracy of spatial discrimination with the whiskers may decline. It needs to test these assumptions through detailed morphological analysis as well as in vivo electrophysiological or imaging experiments to examine neural activity in VPM and S1 neurons in response to whisker stimulation in adolescent SI mice.

We revealed that the plasticity of MLF-VPM synapses in response to SI differs markedly between adolescence and adulthood (Fig. 1, Supplementary

**Fig. 6 | Synapse remodeling by adolescent SI rearing is prevented by deleting GRs from VPM neurons. a–h** Immunohistochemical staining for GR (green), DAPI (blue), and cell-type markers (NeuN in **c, f**, S100B in **d, g**, IBA1 in **e, h**) was performed on male floxed-GR mice (**a, c–e**) and GR-ThKO mice (**b, f–h**) at P30. Asterisks in **a, b** mark the VPM region. **c–h** show higher-magnification views of the VPM. Scale bars: 1 mm (**a, b**), 20 μm (**c–h**). **i** Percentage of GR-positive cells in each cell type (neuron: floxed-GR, N = 3; GR-ThKO, N = 4; df = 5, t = 162.742, ***$p < 0.001$ by Student's $t$-test; astrocyte: floxed-GR, N = 3; GR-ThKO, N = 3; df = 4, $t = 1.190$, $p = 0.300$ by Student's $t$-test; microglia: floxed-GR, N = 3; GR-ThKO, N = 3; df = 4, $t = 0.135$, $p = 0.899$ by Student's $t$-test). Vertical lines indicate mean ± SEM. **j** Male GR-ThKO and floxed-GR mice were weaned at P21 and reared in SI conditions for 4 weeks (floxed-GR, N = 4; GR-ThKO, N = 4). **k** Representative trases of MLF-VPM EPSCs from each condition. **l** Histograms of afferent numbers (floxed-GR, n = 56; GR-ThKO, n = 47; **$p = 0.006$ by $U$-test). **m** Plots of SF- and Max-amplitudes (SF-amplitude: floxed-GR, n = 117; GR-ThKO, n = 76; $p = 0.246$ by $U$-test; Max-amplitude: floxed-GR, n = 56; GR-ThKO, n = 47; $p = 0.146$ by $U$-test). Each point shows the amplitude of an individual fiber and VPM neuron. Scale bars: 5 ms, 1 nA. Vh = −70 mV. Data from different mice are shown using distinct colored symbols (**l, m**). Boxes indicate mean values with error bars showing ± SEM. All data were obtained from male mice.

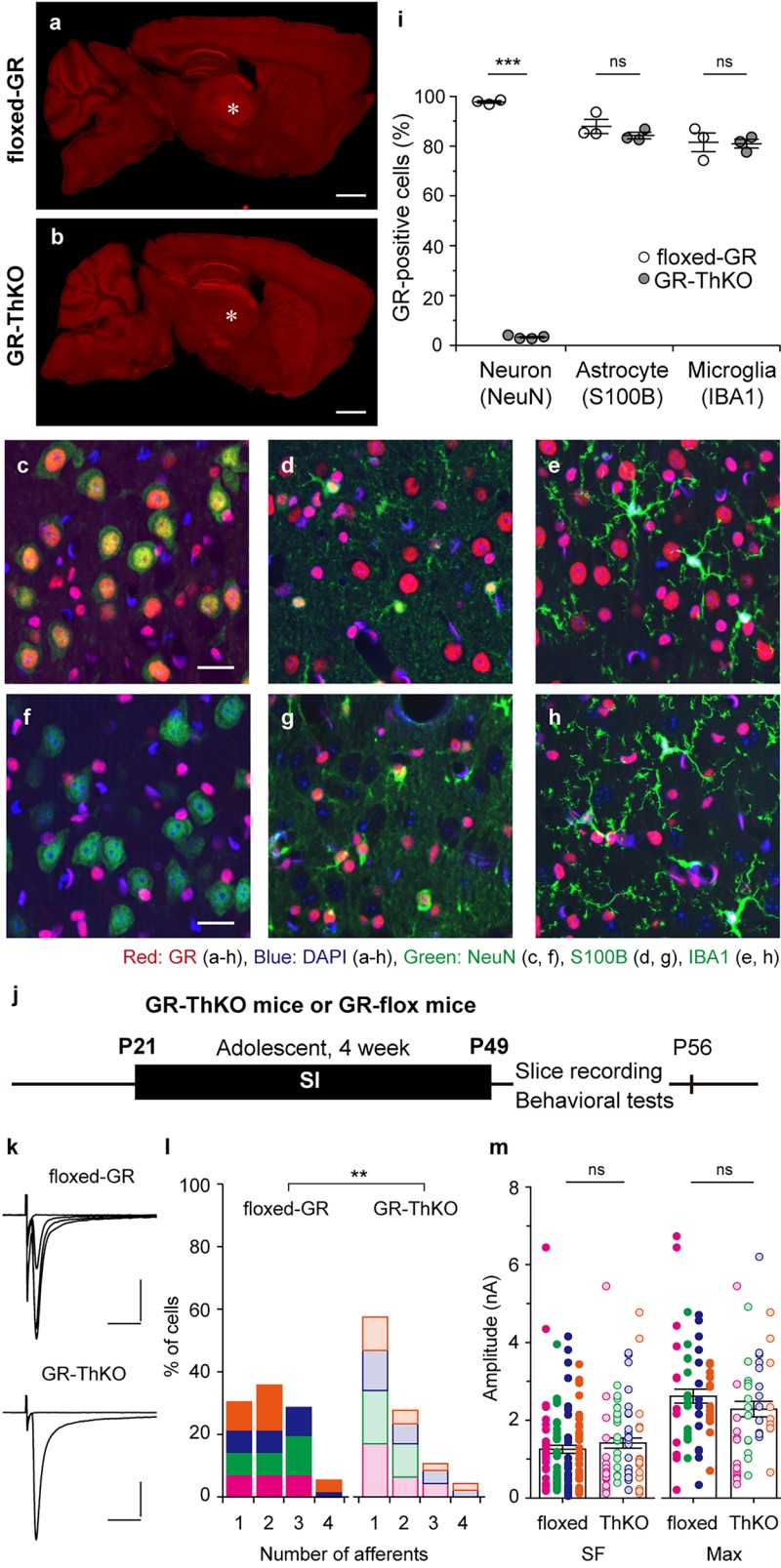

Red: GR (a-h), Blue: DAPI (a-h), Green: NeuN (c, f), S100B (d, g), IBA1 (e, h)

Figs. 3, 4). This raises two possibilities: (1) the effects of SI on the hypothalamic-pituitary-adrenal (HPA) axis may differ across developmental stage, and (2) the MLF-VPM synapse itself may lose its plasticity by adulthood. Our data demonstrate that the same SI paradigm produced different corticosterone dynamics in adolescents and adults (Fig. 4a, b), and that one week of DEX administration in adult mice was sufficient to induce synaptic remodeling (Supplementary Fig. 7). These results suggest that the age-dependent responsiveness of the HPA axis to SI underlies the adolescent-specific synaptic remodeling of MLF-VPM synapses. The responsiveness of the rodent HPA axis to stressors varies across life stages[43–47]. The response of the HPA axis is more prolonged and less habituating in adolescence than in adulthood[48–50], although it depends on the

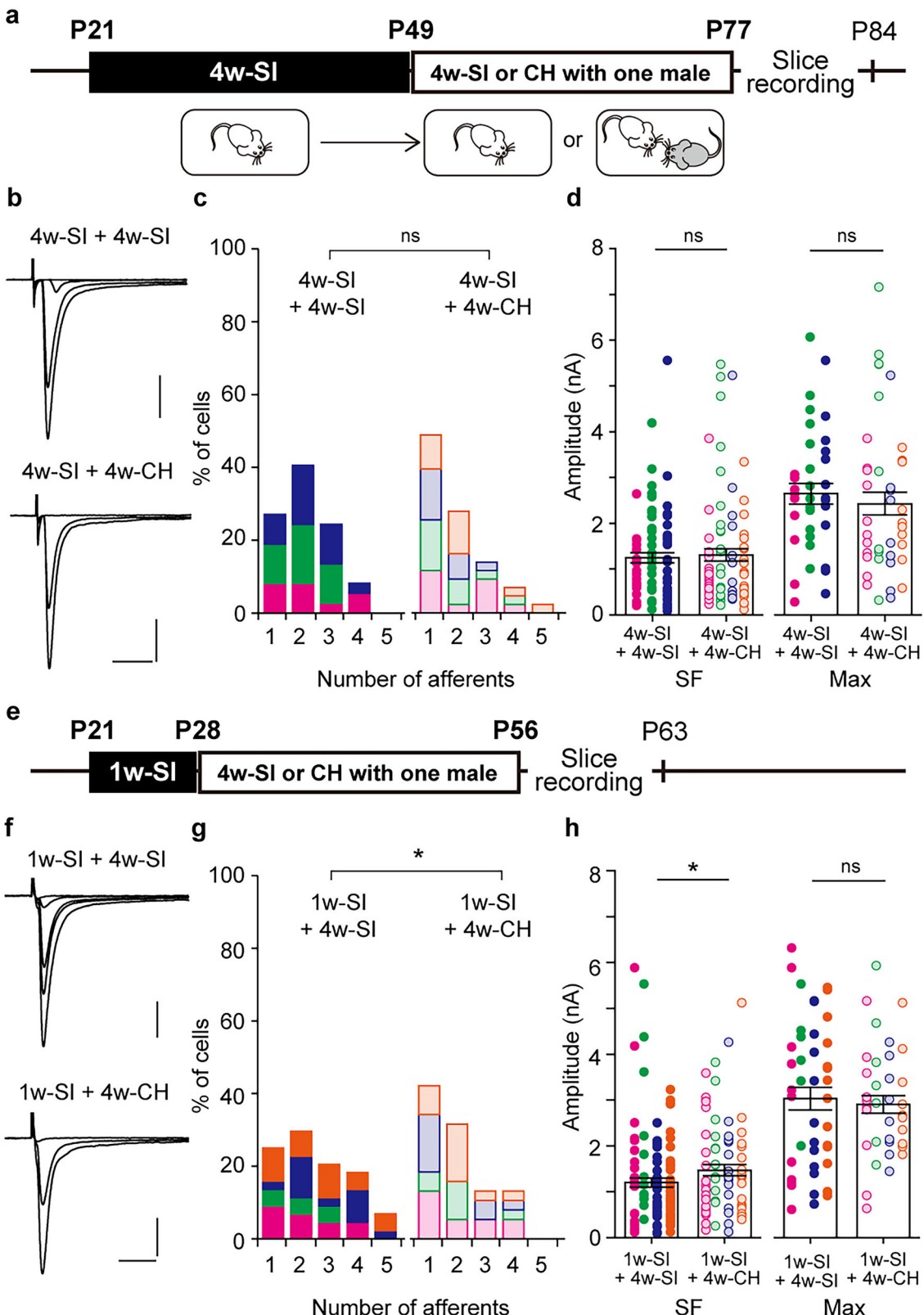

timing within adolescence, the nature of the stressors, and sex[51,52]. Studies in humans report basal salivary cortisol levels and responses to psychosocial stresses such as public speaking are higher in adolescence than in childhood[53]. The intense and prolonged response of the HPA axis in adolescents are thought to reflect immaturity of the inhibitory circuits regulating the HPA system, including networks interconnecting the hippocampus, amygdala, and prefrontal cortex[46,50,54]. Thus, the elevated corticosterone levels observed in adolescent SI may result from the immaturity of the HPA negative-feedback system. Given that the difference in corticosterone dynamics between adolescence and adulthood depends on the nature of stressor[46], it would be interesting to determine what kind of stressor cause stronger HPA activation in adults, and whether such stressor

**Fig. 7 | Remodeling at VPM synapses after a 1-week adolescent SI is partially normalized by 4-weeks of cohabitation with a male GH- mouse. a** Male mice were kept unser SI conditions from P21 for 4 weeks (4w-SI). A male GH mouse was then introduced into SI mouse's cage and cohabited for 4 weeks (4w-CH) (4w-SI + 4w-CH, N = 4; 4w-SI + 4w-SI, N = 3). **b** Representative MLF-VPM EPSCs traces in each condition. **c** Histograms of afferent numbers (4w-SI + 4w-SI, n = 37; 4w-SI + 4w-CH, n = 43; *p* = 0.103 by *U*-test). **d** Plots of SF- and Max-amplitudes (SF-amplitude: 4w-SI + 4w-SI, n = 78; 4w-SI + 4w-CH, n = 80; *p* = 0.776 by *U*-test; Max-amplitude: 4w-SI + 4w-SI, n = 37; 4w-SI + 4w-CH, n = 43; *p* = 0.300 by *U*-test). **e** Male mice were kept in SI conditions for 1 week (1w-SI), then cohabited with a male GH mouse for 4 weeks (4w-CH) (1w-SI + 4w-SI, N = 4; 1w-SI + 4w-CH, N = 4). **f** Representative traces of MLF-VPM EPSCs in each condition. **g** Histograms of afferent numbers (1w-SI + 4w-SI, n = 44; 1w-SI + 4w-CH, n = 38; *p* = 0.042 by *U*-test). **h** Plots of SF- and Max-amplitudes (SF-amplitude: 1w-SI + 4w-SI, n = 111; 1w-SI + 4w-CH, n = 75; *p* = 0.018 by *U*-test; Max-amplitude: 1w-SI+4w-SI, n = 44; 1w-SI+4w-CH, n = 38, *p* = 0.838 by *U*-test). Scale bars: 5 ms, 1 nA. Vh = −70 mV. Each point in **d**, **h** represents the amplitude of individual fiber and VPM neuron, respectively. Boxes represent mean values with error bars showing ± SEM (**d**, **h**). Data from different mice are shown using distinct colors (**c**, **d**, **g**, **h**).

can increase afferent numbers onto VPM neurons and alter whisker-related behaviors. Elucidating the neural mechanisms for age-dependent HPA activation will be beneficial in developing ways to circumvent thalamic synaptic remodeling and changes in brain function under psychosocial stress.

The fact that adolescent SI remodels MLF-VPM synapses shows that thalamic synapses are highly susceptible to environmental influences during adolescence. This implies that it is possible to normalize the remodeled synapses under adolescent SI through appropriate interventions carried out during adolescence, when they exhibit high plasticity. In the present study, we examined the possibility of cohabiting SI mice with an age-matched male GH mouse. We observed a modest yet significant reduction in the afferent number and an increase in SF-amplitude when cohabitation began in adolescence, but not in adulthood; however, the extent of normalization did not match that of mice kept in GH condition (Fig. 7). These results suggest that the effects of adolescent SI rearing, even after only one week, last at least 4 weeks, and full normalization requires additional factors beyond simply cohabiting with another mouse. In our experimental condition, we observed that the 4-week SI mice sometimes attacked cohabitant mice, causing injuries; therefore, cohabitation may have acted as an additional stressor for the SI mice rather than a stress reliever. It has been reported that hippocampus-dependent spatial cognitive dysfunction induced in mice during adolescent SI (P21-P30) is improved by 8 weeks of physical environmental enrichment starting at P30, whereas social interaction deficits and anxiety-like behaviors remain unnormalized[55]. This suggests that physical environmental enrichment contains factors capable of restoring the effects of adolescent SI, but the recovery is not all-or-nothing. Determining whether physical environmental enrichment can reverse SI-induced synapse remodeling remains an issue for future research.

In summary, our findings highlight adolescence as a critical period of heightened stress vulnerability in sensory neural circuits. The GR-dependent remodeling of thalamic synapses observed in our study not only deepens our understanding of how psychosocial stress influences sensory processing, but also suggests that targeted modulation of GR signaling could be a potential strategy to improve stress resilience and well-being during adolescence.

## Methods
### Animals
All experiments were approved by the animal welfare committee of Tokyo Women's Medical University (AE24-065-3-D and GE24-058-3-D). We have complied with all relevant ethical regulations for the use of animals. All experiments were performed using male C57BL/6 N mice, except for Supplementary Figs. 2c, d, and 4. Krox20-Cre mice (*Egr2tm2(cre)Pch*/J, RRI-D:IMSR_JAX:025744) and Ai34D mice in which synaptophysin-tdTomato was expressed in the presence of Cre (B6;129S-*Gt(ROSA)26Sortm34.1(-CAG-Syp/tdTomato)Hze*/J, RRID: IMSR_JAX:012570) were crossed, and the Cre-positive pups (Krox20-Ai34D mice) were used to visualize the MLF synapses originating from the brainstem PrV2, as previously described[21,25]. The floxed-GR mice (C57BL/6-*Nr3c1<tm1.1Himo>*, #RBRC10693)[35] and 5HTT-Cre mice (C57BL/6J-*Tg(Slc6a4-cre)208Ito*, #RBRC10598)[36,37] were provided by the RIKEN BRC (RBRC) through the National BioResource Project of the MEXT, Japan. We generated mice with the knockout of GR in thalamic neurons (GR-ThKO) by intercrossing GR-floxed mice with 5HTT-Cre mice. Genotypes of the mice were examined by PCR of genomic DNA extracted from their somatic tissue. PCR was performed according to the protocol described in the strain data sheet provided on the Jackson Laboratory and RBRC websites. Experiments using transgenic mice (Fig. 6 and Supplementary Fig. 6f, g) and part of behavioral tests (Supplementary Figs. 6b, c, 9) were performed with the mouse genotype and rearing conditions blinded. Others were not conducted blindly.

### Post-weaning rearing conditions
In most experiments, male mice were weaned at P21 and randomly divided into two rearing conditions: "Group-housed (GH)" and "Socially isolated (SI)." Female mice were used in Supplementary Figs. 2c, d, and 4. The GH mice were housed with 2 or 3 siblings of the same sex per cage, while the SI mice were housed individually. In experiments where the two rearing conditions were started at 8 weeks old, mice were weaned at P21, reared in GH condition for 5 weeks, and then divided into GH and SI conditions. In the experiments shown in Fig. 7, the SI mice were cohabited with a male, age-matched GH mouse for 4 weeks. If fighting was observed to the extent that one of the two mice injured the other, the cohabitation was considered stressful, and they were excluded from further experiments. The cages for both conditions were made of transparent polymethyl pentene resin, measuring 170 mm in width, 350 mm in length, and 145 mm in height (#KN-60101; Natsume Seisakusho, Tokyo, Japan), and were covered with an appropriate amount of bedding (ALPHA-dri, Shepherd Specialty Papers, New Hampshire, USA). The SI cages were surrounded by cardboard to prevent visual communication with mice in neighboring cages, but communication via sounds and smells was not restricted. The rearing room was maintained at a temperature of $22 \pm 1\,°C$ and a relative humidity of $50 \pm 15\%$. Lights were on from 8:00 to 20:00. In both rearing conditions, food and water were available ad libitum, and bedding was changed once a week.

### Electrophysiology
Parasagittal brain slices, including the VPM and medial lemniscus fibers (MLFs), were prepared as described previously[19,23,28]. In brief, mice were deeply anesthetized by sevoflurane inhalation and decapitated. The brains were quickly removed, and parasagittal slices with 300 μm thickness were prepared with a vibratome slicer (VT1200S; Leica Microsystems Inc., Wetzlar, Germany) in a chilled sucrose-based cutting solution composed of (in mM) 234 sucrose, 2.5 KCl, 10 $MgCl_2$, 0.5 $CaCl_2$, 11 glucose, 1.25 $NaH_2PO_4$, 25 $NaHCO_3$, and 0.5 Myo-inositol, bubbled with 95% $O_2$ and 5% $CO_2$. After cutting, the slices were first incubated at 35 °C for 20 min in normal artificial cerebrospinal fluid (ACSF) composed of (in mM) 125 NaCl, 2.5 KCl, 2 $CaCl_2$, 1 $MgSO_4$, 1.25 $NaH_2PO_4$, 26 $NaHCO_3$, and 20 D-glucose, bubbled with 95% $O_2$ and 5% $CO_2$, and kept in the ACSF at 25 °C until being moved to a recording chamber. Whole-cell recordings were made from VPM neurons using a near-IR CCD camera (C3077-79; Hamamatsu Photonics, Hamamatsu, Japan) equipped with an upright microscope (Examiner. A1; ZEISS Microscopy, Jena, Germany). Electro-physiological recordings were conducted at 32 °C with an EPC10 patch-clamp amplifier (HEKA Elektronik, Reutlingen, Germany). Signals were filtered at 2.9 kHz and sampled at 50 kHz. The pipette series resistance (Rs) was compensated by 80%. Rs lower than 15 MΩ were permitted for analysis. Online data acquisition and offline data analysis were performed using

PatchMaster and FitMaster software (HEKA Elektronik). The pipette solution was composed of (in mM): 120 CsMeSO$_3$, 20 NaCl, 5 QX314, 2 ATP, 0.5 GTP, 2 MgCl$_2$, 0.1 CaCl$_2$, 10 HEPES, and 1 EGTA (pH 7.3, adjusted with CsOH). MLF-EPSCs were recorded at holding potentials (Vh) of −70 mV and +40 mV. To block inhibitory synaptic transmission, picrotoxin (100 μM; Tocris Bioscience, Bristol, UK) was added to normal ACSF. We electrically stimulated a visually identified bundle of MLFs on the brain slice at 0.2 Hz with a tungsten concentric electrode (UEWMEC-SE2NNM; Frameless Hardware Company, Los Angeles, USA). EPSCs elicited in an all-or-none fashion and showing paired-pulse depression in response to paired stimulation with a 100 ms interval were identified as MLF-EPSCs[20,56,57]. Stimulus intensity was increased in steps of 10 or 20 μA from 0 μA, with 3 to 5 trials at each stimulus intensity. If an EPSC was elicited, we checked whether that EPSC occurred in an all-or-none manner by varying the stimulus intensity around the initial value. The upper limit of stimulus intensity was twice the threshold intensity of the last evoked EPSC or 200 μA, whichever was greater (Supplementary Fig. 1a, b). The number of MLFs innervating the recorded VPM neuron was estimated as the number of discrete EPSC steps in response to gradually increasing stimulus intensity (Supplementary Fig. 1a, b, c). Since previous studies have shown that the miniature EPSC of the MLF-VPM synapse at >P21 is approximately 25 pA[19], the lower limit of the EPSC step size in the present study was set at 50 pA. The single-fiber (SF-) amplitude was calculated as the amplitude difference between steps, and the sum of the single-fiber amplitudes per cell was calculated as the maximum (Max-) amplitude (Supplementary Fig. 1a, b, d, e). The paired-pulse ratio (PPR) was calculated in EPSCs with the smallest stimulus intensity in each cell, using the ratio of the peak amplitude in the second response to the first response (Supplementary Fig. 1f).

### Whisker deprivation

Bilateral mystical vibrissae on the snout of mice were carefully removed every 3 days for 4 weeks, starting at weaning at P21. The procedure was performed under anesthesia with isoflurane (4% for induction, 1.5–2.0% for maintenance) and under a dissecting microscope. Control mice only underwent anesthesia, matching those in the whisker-deprived condition.

### In vivo drug application

The process of in vivo drug application involved the intraperitoneal injection of dexamethasone (DEX, 5 mg/kg body weight, #1126; Tocris Bioscience) and mifepristone (RU486, 40 mg/kg body weight, #M8046; SIGMA-ALDRICH, St. Louis, USA) once a day for the first week after weaning. The same volume of vehicle sesame oil (VHCL, #S3547; SIGMA-ALDRICH) was injected as a control.

### Immunohistochemistry

Mice were deeply anesthetized by sevoflurane inhalation, and perfusion fixation with 4% paraformaldehyde fixative supplemented with 0.2% picric acid was conducted. Brains were removed, postfixed in the same fixative at 4 °C overnight, and stored in phosphate-buffered saline (PBS). Parasagittal sections of 50 μm thickness were prepared with a vibratome slicer (VT1000S; Leica Microsystems Inc.). The slices were incubated overnight at 4 °C in 10% normal donkey serum, followed by incubation with the primary antibodies. The primary antibodies used in the present study were guinea pig anti-VGluT2 (1:500, VGluT2-GP-Af810; Frontier Institute, Sapporo, Japan), rabbit anti-glucocorticoid receptor (1:2000, ab183127; Abcam, Cambridge, UK), chicken anti-S100B (1:500, 287 006; Synaptic Systems, Goettingen, Germany), goat anti-IBA1 (1:500, 011-27991; FUJIFILM Wako Pure Chemical Corporation, Osaka, Japan), and mouse anti-NeuN (1:500, MAB377; Merck, Darmstadt, Germany). After incubation with the primary antibodies, sections were washed and incubated with Alexa Fluor 405-, 488-, and 568-conjugated species-specific secondary antibodies (1:500; Thermo Fisher Scientific, Waltham, USA) for two hours at room temperature. Sections stained with GR and NeuN, S100B, or IBA1 were mounted with DAPI-containing media (ProLong Diamond Antifade Mountant with DAPI, P36962, Thermo Fisher Scientific).

For imaging of the entire brain sections (Fig. 6a, e, Supplementary Fig. 6a), an all-in-one fluorescence microscope (BZ-X800, KEYENCE, Osaka, Japan) equipped with an advanced observation module (BZH4XD, KEYENCE) and a Plan Apochromat 20x objective (NA 0.75, BZ-PA20, KEYENCE) was used to capture tiled images. The images were stitched with the BZ-X Analyzer software (BZ-H4A, KEYENCE). Images were obtained with an all-in-one fluorescence microscope (BZ-X800, KEYENCE, Osaka, Japan). Images of the VPM region were obtained by Z-stacking 16 images, taken every 0.3 μm, using the BZ-X800 microscope equipped with a Plan Apochromat 40x objective (NA 0.95, BZ-PA40, KEYENCE) and an optical sectioning module (BZ-H4XF, KEYENCE).

The experimental procedure for quantifying the percentage of cells expressing GR for each cell type is as follows (Fig. 6b–d, f–h, Supplementary Fig. 8c, f–n). First, structures above a certain fluorescence intensity and larger than 5 μm$^2$ in size for each cell marker were extracted from the Z-stacked images of cell markers (NeuN, S100B, IBA1). Unseparated structures were manually separated as needed. Next, DAPI-positive structures among them were further extracted. Finally, GR-positive structures were further extracted among the cell marker- and DAPI-positive structures. The ratio of cells expressing GR per cell type was calculated as the number of cell marker-, DAPI-, and GR-positive structures divided by the number of cell marker- and DAPI-positive structures (Figs. 6b–d, f–h, Supplementary Fig. 6c, f–n). We analyzed 3 or 4 images of the VPM, each 369 × 277 μm, per mouse, and their averaged values were used as the data for each mouse. The extraction of positive areas for each antibody was calculated using a hybrid cell count application (BZ-H4C, KEYENCE) in the BZ-X Analyzer software (BZ-H4A, KEYENCE).

The procedure for analyzing the co-localization of VGluT2 and tdTomato was also performed using the same software as above. First, structures above a certain fluorescence intensity and 1 μm$^2$ in size on the VGluT2-stained image were extracted. If necessary, fused structures were separated manually. Among the extracted VGluT2-positive structures, those with tdTomato signal areas above a particular signal intensity were extracted, and the number of VGluT2-positive structures and the number of VGluT2 and tdTomato-positive structures were obtained. We analyzed 3 or 4 images of the VPM per mouse, each measuring 369 × 277 μm, and their averaged values were used as data for each mouse (Supplementary Fig. 5a).

### Behavioral tests

Mice cages were moved to the behavioral room two days before the tests to acclimate the mice to the environment. Tests were conducted between 10:00 and 16:00. To assess whisker-related somatosensory function of GH and SI mice, we performed a tactile object recognition test (tNORT)[29–32] and gap-cross test[34,58]. The mice's behavior during the test was recorded with a CCD camera (CM3-U3-13S2M-CS, FLIR) at 30 frames/s for the tNORT and 15 frames/s for the gap-cross test.

In tNORT, trajectories of the center and nose tip of the mice were measured using video tracking software (ANY-maze, Stoelting, Wood Dale, USA). Behavioral tests were composed of habituation (open field test) on days 1 and 2, tNORT on day 3, and NORT on day 4. In the habituation sessions, mice were placed in an empty, gray-colored chamber (50 cm × 50 cm × 50 cm) and allowed to explore the arena freely for 10 min each day. The distance the center of the mice moved (total distance) and the time spent in the center of the arena (30 cm × 30 cm, center time) during the habituation on day 1 were measured (Fig. 2b–d). On day 3, tNORT was conducted in the same arena used for habituation. Following a study by Kwak and colleagues[31], we prepared three types of objects (cylinders with 2 cm diameter and 9 cm high) wrapped in sandpaper with distinct roughness (400-, 240-, and 60-grit, designated as G400, G240, and G60, respectively). The tNORT consisted of three sessions: learning, test 1, and test 2. During the learning session, mice were placed in an arena with two G400s (G400 and G400') and allowed to explore freely for 5 min. The mouse was moved in a transporting cage. While the mice were waiting in the transfer cage, the G400' was replaced with G240, and the mice were returned to the arena to explore for 5 min (test 1 in Fig. 2). The mice were again allowed to wait in the transfer cage. During that time, the

G240 was replaced with G60, and the mice were returned and allowed to explore for 5 min (test 2 in Fig. 2). The presentation positions of G400' and G400 were swapped for each mouse. The time the mouse's nose tip was within 1.5 cm of each object was measured in each session and used to calculate the discrimination index as follows. Discrimination index in the learning session was calculated as time (G400') – time (G400)/total exploration time, and discrimination index in test1 and test2 as time (G240 or G60) – time (G400)/total exploration time (Fig. 2e, g). The higher discrimination index indicates that the mouse spends more time exploring the new object than the familiar one. On day 4, NORT was performed in the same arena as days 1 to 3. The NORT procedure was identical to tNORT except for the objects presented: a black cube (3 cm × 3 cm × 3 cm) and a white ball (3 cm diameter). Mice first explored the arena where the two black cubes ('Cube and Cube') were placed for 5 min during the learning session. Then, after a 5-min waiting period, mice explored an arena with a black cube and a white ball for 5 min in test 3. The presentation positions of Cube' and Cube were swapped for each mouse. Discrimination index in the learning session was calculated as time (Cube') – time (Cube)/total exploitation time, and discrimination index in test 3 as time (Ball) – time (Cube)/total exploitation time (Fig. 2f, h). In tNORT and NORT, objects were placed at equal distances from the center of the arena and the wall. Between sessions and at the end of each mouse trial, the object and arena were cleaned with 70% ethanol and distilled water to remove olfactory cues. In the analysis of tNORT and NORT, we excluded data from mice that explored two objects for less than 2 s or explored only one object in any session. However, total distance and center time in habituation include data from all mice tested.

In the gap-cross test, a small, unenclosed starting platform (7 cm wide, 8 cm long) and a platform (7 cm wide, 20 cm long) covered on both sides by opaque walls 20 cm high were used. The two platforms were placed at a height of 20 cm, and the distance between them was varied. Initially, a mouse was habituated to the platforms, arranged with no gaps between them, for 5 min. The gap between the platforms was then increased by 1 or 0.5 cm, and mice were placed on the starting platform to see if they could cross the gap to the opposite platform. We measured the maximum gap distance they could cross within five minutes. The experiment was terminated after two consecutive trials in which the mouse explored the opposite platform with whiskers but did not attempt to cross the gap, or failed to cross the gap. Mice that did not explore the opposite platform during the test session and mice that repeatedly jumped off the platform were excluded from the analysis.

von Frey test was performed to assess mechanical sensitivity[24,26–28]. The experimenter gently held the mice in her hand. After the mice calmed down and remained still in the experimenter's hands, von Frey filaments (Aesthesio, Danmic Global, Campbell, USA) were applied to the whisker pad with increasing bending forces from 0.008 to 20 g. Tests were conducted six times for each bending force of the filament. When mice showed a withdrawal response to mechanical stimuli more than four times, the force of that filament was defined as the withdrawal threshold. The series was repeated three times, and the mean values obtained were used for statistical comparisons. Values that were more than three times greater than or less than one-third of the average of the other two values were considered outliers and excluded from the calculation of each mouse's threshold. In our experiments, we did not habituate the animals to being held in the experimenter's hands beforehand to prevent possible effects of tactile stimuli during handling on MLF-VPM synapses.

## Urinal corticosterone measurement

Spot urine was collected between 10:00 and 11:00 every 7 days, and corticosterone levels were measured with the Corticosterone EIA Kit (#YK240; Yanaihara Institute Inc., Fujinomiya, Japan). The authors collected samples, and measurements were carried out using commercial research support services (Yanaihara Institute Inc.).

## Statistics and reproducibility

All experimental conditions include a minimum of three mice. The number of samples and the number of mice used to obtain them are noted in the figure legend as n and N, respectively. The sample size was determined based on previous studies conducted in similar analyses[19,23,28,59]. Throughout the figures, data for each mouse are presented differentiated by color and fill, and group means are presented with ± SEM. The number of samples used for analysis in each experiment is denoted by 'n' and the number of mice by 'N' in the text and legend. Group comparisons were performed using Student's t-test when data satisfied both normality (Shapiro–Wilk test) and equal variance (Brown–Forsythe test). If normality was not met, the Mann–Whitney U-test was used and is referred to as 'U-test' throughout the manuscript. A paired t-test was applied for the NORT data (Fig. 2f, h, Supplementary Fig. 9h) after confirming the dataset passed the Shapiro–Wilk test. In the tNORT (Fig. 2e, g, Supplementary Fig. 9g), a one-way repeated measures ANOVA, followed by a post-hoc test of Holm-Sidak multiple comparisons, was used after confirming the assumptions of normality and sphericity using the Shapiro–Wilk test and Mauchly's test, respectively. The same statistical method was used for the same parameters throughout the experiment. Statistically significant differences were set at *$p < 0.05$, **$p < 0.01$, and ***$p < 0.001$. 'ns' indicates nonsignificant differences. Statistical analyses were conducted using the SigmaPlot 14.5 program (Systat Software Inc., San Jose, CA, USA).

## Reporting summary

Further information on research design is available in the Nature Portfolio Reporting Summary linked to this article.

## Data availability

The datasets generated during and/or analyzed during the current study are available from the corresponding author upon reasonable request. The numerical source data for the figures were provided as Supplementary Data 1.

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

## Acknowledgements

We thank Sachie Sekino for conducting behavioral experiments; the Institute of Laboratory Animals of Tokyo Women's Medical University (TWMU) for animal care; the RIKEN BRC through the National BioResource Project of the MEXT/AMED for providing floxed-GR mice and 5HTT-Cre mice; Prof. Takuji Iwasato (National Institute of Genetics) for kindly supplying Krox20-Cre mice; and the Gunma University Initiative for Advanced Research Viral Vector Core for kindly packaging and purifying viral particles. We thank Edanz (https://jp.edanz.com/ac) for their assistance in editing an early draft of this manuscript. This work was supported by JSPS KAKENHI [JP20K06862 and JP23K06813 to H.N., 23H02592 and 23K18258 to M.M.]; Grant-in-Aid for Transformative Research Areas (JP20H05916 to M.M.); the AMED program of Brain/MIND (JP19dm0207057 to the Gunma University Initiative for Advanced Research Viral Vector Core); Brain Science Foundation to H.N.; and a research grant from TWMU Career Development Center for Medical Professionals to H.N.

## Author contributions

H. N. and M. M. designed the research. H. N. performed the experiments and analyzed the data with the help of Sachie Sekino, a technician in our laboratory. H. N. and M. M. wrote the paper.

## Competing interests

The authors declare no competing interests.
