## [Transparent Peer Review file · Communications Biology]

Adolescent stress remodels synapses in the sensory thalamus and impairs tactile discrimination in mice.

Corresponding Author: Dr Hisako Nakayama

Version 0:

Reviewer comments:

Reviewer #1

(Remarks to the Author)

In the current manuscript, the authors tested whether adolescent social isolation impacts neural circuits in the somatosensory thalamus. They first demonstrated that social isolation during adolescence, but not adulthood, remodels MLF-VPM synapses from single- to multiple-innervation, as well as leads to hypersensitization of tactile perception and altered whisker-dependent cognitive behavior. Furthermore, they revealed a significant elevation in urinary corticosterone uniquely following adolescent social isolation, which interestingly plays a causal role in the altered remodeling of MLF-VPM synapses induced by adolescent social isolation. Additionally, glucocorticoid receptors in VPM neurons were responsible for the remodeling of MLF-VPM synapses induced by adolescent social isolation. Given that pharmacological administration of DEX during the first week of adult social isolation also induced synapse remodeling, the authors concluded that the adolescent-specific effect of social isolation on MLF-VPM synaptic remodeling could be explained by distinct responses to psychosocial stress rather than differences in neural plasticity across developmental periods. Overall, findings are novel and provide important mechanistic insights into the effect of early social isolation. The authors could benefit from addressing the following points;

Major comments:

- 1) This manuscript lacks quantification of the immunohistochemical experiments (Fig. 6a-d, Extended Fig. 4a-b, Extended Fig. 5). In addition to describing the observations, the authors should provide a discussion of the results based on quantification.
- 2) In the last paragraph of the Results, the manuscript includes additional discussion, primarily about the data in Fig. 5a-d. Since this paragraph does not introduce new main figure data, combining it with the paragraph on lines 238–287 could improve the flow of the paper, or alternatively, it could be moved to the Discussion section.
- 3) In lines 482–485, the authors mention that re-housing after adolescent social isolation normalizes the behavioral and neural abnormalities induced by social isolation, citing reference 77 (Makinodan et al., 2017). However, the rescue effects differ depending on the re-housing condition (i.e., whether the animals were rehoused with group-housed or isolated mice) (for example, see Makinodan et al., 2012; Yamamuro et al., 2020; Bicks et al., 2020). To avoid a misleading interpretation, please describe the re-housing conditions more carefully.

References:

10.1126/science.1220845
10.1038/s41593-020-0695-6
10.1038/s41467-020-14740-z

Minor comments:

- 1) In line 93 of the introduction, the MLF-VPM is mentioned without any prior explanation. It would be more helpful to provide an explanation of the MLF in the introduction, rather than waiting until line 108 of the Results section to do so.

- 2) Lines 132-3: data on PPR should be presented as a supplemental figure.
- 3) The authors include the data showing differences in body weight between groups. Is this because the body weight could impact the tactile sensations or its related behaviors?
- 4) In lines 365, Please confirm if "However" is a correct conjunction or a misdescriptions.
- 5) Line 683 Figure 2 legend: Caling gap-corssing task as sensory-"cognitive" behavior is a bit of stretch. I sugges to rephrase.

Reviewer #2

(Remarks to the Author)

This study examines how early-life social isolation affects sensory processing and synaptic remodeling in the somatosensory system, linking these changes to altered corticosterone levels. While the topic is relevant to neurodevelopmental disorders, key methodological issues weaken the conclusions.

The study does not properly account for sex differences, nor does it assess whether effects persist into adulthood, limiting its developmental relevance. Methodological flaws, such as the use of manual restraint in the Von Frey test and unaccounted body weight differences in the Gap Cross Test, further compromise data reliability. Additionally, the link between corticosterone and synaptic remodeling remains speculative without confirming peak amplitude changes in adulthood.

Below are specific comments addressing these concerns in detail.

Abstract.

Lines 30 to 35: The sentence is too long and difficult to follow. Please rephrase it for clarity.

Introduction.

Line 93: The acronym "MLS" needs to be spelled out the first time it is introduced. What does it stand for?

Line 98: Similarly, define "GR" at first mention.

Results.

Line 106 to 108: A reference must be added to support this statement.

Line 106 to 114: This section does not belong in the results; it is background information and should be moved to the introduction. Additionally, expand the description of sensory system organization in relation to whiskers.

Line 117 to 122: This content pertains to the methods rather than the results. Expand it and integrate it into the methods section with a dedicated paragraph.

Line 129: Add "condition" to "SI" for clarity.

Lines 138 to 140: To properly assess or rule out sex differences, male and female mice should be directly compared for single-fiber amplitudes. It is not sufficient to state that the difference is reduced—this must be supported by statistical testing.

Lines 143 to 145: Sex differences should also be examined at later developmental stages, as they may emerge due to sex maturation.

Lines 148 to 153: The authors must include a comparison of male and female animals in adulthood. Psychiatric conditions such as autism spectrum disorders exhibit sex differences at molecular and physiological levels, which may be more pronounced at specific developmental stages. The absence of differences in early development does not rule out their emergence later.

It is unclear how many animals were used for the experiments in Figure 1 and Extended Figure 2. The authors must clarify this.

Lines 188 to 190: The methodology used for the Von Frey test is deeply flawed. Manually restraining mice during the test introduces significant stress, which can alter sensory thresholds and render the data unreliable. Standard procedures require animals to be tested in a containment chamber after an acclimation period to minimize stress and obtain valid reflexive responses. Without these methodological corrections, the results from this dataset are invalid. The authors must revise their approach using a standardized protocol. Otherwise, the conclusions drawn from this test cannot be supported.

Lines 197 to 199: The use of the Gap Cross Test in this study is highly problematic. The experimental group had significantly higher body weight than controls during adolescence, which could have influenced the results. Since body weight directly affects motor performance and balance, the observed differences cannot be solely attributed to sensory or whisker-dependent factors but may be confounded by biomechanics. Unless body weight differences are accounted for, the conclusions drawn from this test remain unreliable. The authors must normalize for weight differences or reconsider their interpretation of the data. This is further supported by the fact that, in adulthood, when SI mice have a body weight similar to GH mice, no difference in gap distance is observed.

To improve behavioral assessment, I strongly recommend the inclusion of a textured novel object recognition test (PMID: 39025432; PMID: 23026377) or a whisker nuisance test (PMID: 37691105; PMID: 34791077), as these are more reliable measures of whisker sensitivity.

Lines 240 to 242: The statement here is an overstatement. Please revise it to accurately reflect the data.

Line 242 to line 247: urinary corticosterone levels were measured at different time points in only 4 animals per condition. One of the animals for the SI condition (figure 4a) seems an outlier at day 7 (the one that shows corticosterone levels of almost 150ng/ml), also in consideration that the other 3 animals have corticosterone levels ranging between 60 and 90ng/ml. this is also crucial in consideration of the implication of the results the authors claim as they selected the 1 week time point for successive analysis.

The very same can be found for adult SI mice on day 21 (figure 4b).

It is not clear how many animals they have used for experiments shown in 4C-F.

Line 252 to line 259: A fundamental confirmatory experiment should also assess peak amplitude in adulthood at P28. If the authors claim that corticosterone has an effect on synaptic remodelling, they should also see the effect in adult animals when corticosterone levels are higher.

Reviewer #3

(Remarks to the Author)

This manuscript was reviewed through a joint effort by me and my postdoc, Mingyu Yang (mingyu.yang@uk-koeln.de)

Version 1:

Reviewer comments:

Reviewer #1

(Remarks to the Author)

The revised manuscript addressed my concerns with the original manuscript. There was one typo in line 274 "apmlitude-> amplitude.

Reviewer #2

(Remarks to the Author)

I thank the authors for thoroughly addressing my comments. I believe the manuscript has significantly improved and will be of interest to the journal's broad readership.

Reviewer #3

(Remarks to the Author)

The authors have answered to most of my comments and I do think the manuscript has significantly improved (although it would benefit from an extensive proofreading to fix typos; see comments below). However, the authors haven't answered to one of my most important concerns regarding the GR-ThKO mouse line which does not seem to be "thalamic neuron-specific" as the authors claim. If I understand correctly, this is a general and not area-specific KO mouse which lacks GR in serotonin (5HTT-positive) fibers across brain areas (raphe nuclei, multiple thalamic and cortical areas etc) and not exclusively in the VPM. This makes interpretation of the behavior results using this mouse line inconclusive. Can the authors explain in detail how the GR expression in this mouse line is specific to the thalamus? If this is not the case, I'd again suggest to (1) change genetic strategy with an area-specific deletion of GR in the VPM, or (2) use the GR-ThKO mouse only in the slice experiments without the behavior.

Remaining points:

1. The authors discussed the influence of cohousing a male GH mouse with a male SI mouse. They showed that group housing after one week of social isolation did not fully restore MLF-VPM connectivity to GH levels. They suggest that introducing another male to an SI mouse may itself be a stressor, given that they observed attack behavior. This raises a follow-up question: what would be the cohabiting effect in female mice, who are generally less aggressive toward newly-introduced females?

2. Statistics: : Line 849, it should be "Welch's t-test" instead of "Wheich's t-test." For one-way repeated measures ANOVA, testing homogeneity of variance (e.g., Brown-Forsythe test) is not required. Instead, sphericity must be tested, and if violated, corrections such as Greenhouse-Geisser should be applied. In general, it is not accepted to replace a two-way repeated measures ANOVA with Mann-Whitney U tests. If ANOVA assumptions are strongly violated, data transformation should be attempted first.

3. Supplementary Figure 1a & 1b: Please provide a description of the color code used.

Text & typos:

1. The sentence between lines 78-79 describes the study's findings, which fits better in the following paragraph (lines 80-90). Please either revise this sentence or move it to the next paragraph to improve the flow.

2. Post-weaning rearing condition: The first sentence (Lines 624-625) is somewhat confusing; please rephrase for clarity.

3. Line 117: The N for the number of GH afferents is missing. Please recheck the cell number for the SI group; currently it is

written as $n = 3$.

4. Line 123: Please replace "SH" with "SI."
5. Line 152: Typo — should be "after" instead of "afater."
6. Line 295: Typo — should be "difference" instead of "dirrerence."
7. Line 304: Typo — should be "recordings" instead of "recoedings."
8. Line 528: Typo — should be "form" instead of "from."
9. Line 897: Typo — should be "arena" instead of "area."

Version 2:

Reviewer comments:

Reviewer #3

(Remarks to the Author)

I thank the authors for addressing my comments. I believe the manuscript has significantly improved.

Responses to reviewers:

We sincerely thank the reviewers for their positive feedback and helpful suggestions on our manuscript. We have done our best to address their comments and followed their suggestions as far as possible. We believe our manuscript has undergone significant improvement. We have highlighted the changes in the main text and figure legends with light blue markers to make them easily identifiable.

Responses to Reviewer #1:

(Reviewer's comment)

In the current manuscript, the authors tested whether adolescent social isolation impacts neural circuits in the somatosensory thalamus. They first demonstrated that social isolation during adolescence, but not adulthood, remodels MLF-VPM synapses from single- to multiple-innervation, as well as leads to hypersensitization of tactile perception and altered whisker-dependent cognitive behavior. Furthermore, they revealed a significant elevation in urinary corticosterone uniquely following adolescent social isolation, which interestingly plays a causal role in the altered remodeling of MLF-VPM synapses induced by adolescent social isolation. Additionally, glucocorticoid receptors in VPM neurons were responsible for the remodeling of MLF-VPM synapses induced by adolescent social isolation. Given that pharmacological administration of DEX during the first week of adult social isolation also induced synapse remodeling, the authors concluded that the adolescent-specific effect of social isolation on MLF-VPM synaptic remodeling could be explained by distinct responses to psychosocial stress rather than differences in neural plasticity across developmental periods. Overall, findings are novel and provide important mechanistic insights into the effect of early social isolation. The authors could benefit from addressing the following points;

(Our response to Reviewer #1)

We sincerely thank Reviewer #1 for thoroughly reviewing our manuscript and providing valuable suggestions and essential comments. In response to their feedback, we revised our manuscript with additional experiments and analyses. Responses to individual comments are provided below.

Major comments:

(Reviewer's comment)

1) This manuscript lacks quantification of the immunohistochemical experiments (Fig. 6a-d, Extended Fig. 4a-b, Extended Fig. 5). In addition to describing the observations, the authors should provide a discussion of the results based on quantification.

(Our response to the comments)

We indeed thank Reviewer #1 for pointing out this critical issue. We have quantified the

percentage of GR expression in neurons, astrocytes, and microglia in the thalamic VPM of WT, floxed-GR, and GR-ThKO mice, and presented the results with images (Lines 361-367, 373-379, Fig. 6a-l, Supplementary Fig. 8). Additionally, the analysis procedures are described in detail in the Methods section (Lines 712-718, 724-747).

(Reviewer's comment)

2) In the last paragraph of the Results, the manuscript includes additional discussion, primarily about the data in Fig. 5a-d. Since this paragraph does not introduce new main figure data, combining it with the paragraph on lines 238–287 could improve the flow of the paper, or alternatively, it could be moved to the Discussion section.

(Our response to the comments)

We thank Reviewer #1 for this valuable suggestion. Following the advice, we have combined the description for Supplementary Fig. 7 (DEX effects on adult MLF-VPM synapses) with the results of Fig. 5 (DEX effects in adolescent MLF-VPM synapses) (Lines 342-354).

(Reviewer's comment)

3) In lines 482–485, the authors mention that re-housing after adolescent social isolation normalizes the behavioral and neural abnormalities induced by social isolation, citing reference 77 (Makinodan et al., 2017). However, the rescue effects differ depending on the re-housing condition (i.e., whether the animals were rehoused with group-housed or isolated mice) (for example, see Makinodan et al., 2012; Yamamuro et al., 2020; Bicks et al., 2020). To avoid a misleading interpretation, please describe the re-housing conditions more carefully.

References:

10.1126/science.1220845

10.1038/s41593-020-0695-6

10.1038/s41467-020-14740-z

(Our response to the comments)

We thank Reviewer #1 for highlighting the critical points and for providing references. In the revised manuscript, we experimentally verified whether reorganized synapses can be normalized by cohabiting adolescent 4-week or 1-week SI mice with one GH and an age-matched male mouse for 4 weeks (Fig. 7). The results are mentioned in lines 427-455 and discussed in lines 574-591. Shortly, we found a mild but significant normalization in the afferent number and single-fiber amplitude when adolescent 1w-SI mice were cohabited with a GH-mouse. However, compared to the adolescent 4w-GH mice shown in Fig. 1a-d, the number of afferents in the 1w-SI+4w-cohabitation (CH) was significantly higher ($p = 0.003$), and the amplitude of single fibers was smaller ($p = 0.005$). These results suggest that full normalization requires additional factors beyond adolescent CH. Elucidating these factors remains a future challenge.

Minor comments:

(Reviewer's comment)

1) In line 93 of the introduction, the MLF-VPM is mentioned without any prior explanation. It would be more helpful to provide an explanation of the MLF in the introduction, rather than waiting until line 108 of the Results section to do so.

(Our response to the comments)

We have added an explanation of MLF to the introduction (Line 68).

(Reviewer's comment)

2) Lines 132-3: data on PPR should be presented as a supplemental figure.

(Our response to the comments)

Following this advice, we presented the PPR data as Supplementary Figs. 1 and 2.

(Reviewer's comment)

3) The authors include the data showing differences in body weight between groups. Is this because the body weight could impact the tactile sensations or its related behaviors?

(Our response to the comments)

We appreciate Reviewer #1 for highlighting an important point. As Reviewer #1 pointed out, adolescent SI had accelerated weight gain compared to the GH group. While the fact that the difference in the score of gap-cross tests between GH and SI disappeared after the removal of whiskers (with whiskers: $p=0.001$; without whiskers: $p=0.284$) supported that the poor performance of SI mice depends on whiskers, we cannot completely rule out the possibility that the weight gain may affect the score with other factors. Therefore, we revised our assessment to evaluate the effect of whisker deprivation on the scores of gap-cross tests under each rearing condition, rather than a direct comparison of GH and SI (Lines 242-261), and relocated the data to Supplementary Fig. 6a, c, d, and f. Additionally, the results of the von Frey tests were relocated to Supplementary Fig. 6b,e, as another reviewer raised critical concerns that restraining mice during the test could alter their emotional state and behavior. The deficiencies of the experimental method are explained in the Results section (Lines 191-197), along with the data. In the revised manuscripts, we conducted the tactile novel object recognition test (tNORT), which has been proven in previous studies to evaluate the tactile sense of whiskers (Wu et al., 2013, Behav Brain Res, DOI: 10.1016/j.bbr.2012.09.044). The results are mentioned in lines 198-241 and Fig. 2. The tNORT demonstrated that mice exposed to adolescent SI showed decreased tactile discrimination compared to adolescent GH mice.

(Reviewer's comment)

4) In lines 365, Please confirm if "However" is a correct conjunction or a misdescriptions.

(Our response to the comments)

As in the response to comment (3), we revised the section on behavioral experiments (Lines 182-265).

(Reviewer's comment)

5) Line 683 Figure 2 legend: *Caling gap-corssing task as sensory-"cognitive" behavior is a bit of strech. I sugges to re-phrase.*

(Our response to the comments)

We removed the term 'sensory-cognitive behavior' and instead referenced the tNORT and gap-cross test as 'whisker-related behavior' in the revised manuscripts.

Responses to Reviewer #2:

(Reviewer's comment)

This study examines how early-life social isolation affects sensory processing and synaptic remodeling in the somatosensory system, linking these changes to altered corticosterone levels. While the topic is relevant to neurodevelopmental disorders, key methodological issues weaken the conclusions.

The study does not properly account for sex differences, nor does it assess whether effects persist into adulthood, limiting its developmental relevance. Methodological flaws, such as the use of manual restraint in the Von Frey test and unaccounted body weight differences in the Gap Cross Test, further compromise data reliability. Additionally, the link between corticosterone and synaptic remodeling remains speculative without confirming peak amplitude changes in adulthood.

Below are specific comments addressing these concerns in detail.

(Our response to Reviewer #2)

We sincerely appreciate Reviewer #2 for thoroughly reviewing our manuscript and providing valuable suggestions and essential comments. According to the comments of Reviewer #2, we revised our manuscript with additional experiments and analysis. Responses to individual comments are as follows.

(Reviewer's comment)

Abstract.

Lines 30 to 35: The sentence is too long and difficult to follow. Please rephrase it for clarity.

(Our response to the comments)

We thank Reviewer #2 for this advice. Following the suggestions of other Reviewers, we have revised the abstract and introduction to align with the experimental content and removed the relevant sentences. Please review the revised abstract (Lines 30-42) and introduction (Lines 45-90).

(Reviewer's comment)

Introduction.

Line 93: The acronym "MLS" needs to be spelled out the first time it is introduced. What does it stand for?

Line 98: Similarly, define "GR" at first mention.

(Our response to the comments)

Abbreviations were mentioned with full names when they first appeared (MLF: line 68, GR:

line 35).

(Reviewer's comment)

Results.

Line 106 to 108: A reference must be added to support this statement.

(Our response to the comments)

We added references (Kawasaki, 2015, Dev Growth Differ, DOI: 10.1111/dgd.12208; Iwasato and Erzurumlu, 2018, Curr Opin Neurobiol, DOI: 10.1016/j.conb.2018.06.001; Lines 64-67).

(Reviewer's comment)

Line 106 to 114: This section does not belong in the results; it is background information and should be moved to the introduction. Additionally, expand the description of sensory system organization in relation to whiskers.

(Our response to the comments)

Following the advice, we have moved this section to the introduction (Lines 64-79). In addition, the structural feature of the conduction pathway from the whiskers to the somatosensory cortex, where the arrangement of the whiskers is reproduced in each relay nucleus, was described in the discussion section (Lines 509-514).

(Reviewer's comment)

Line 117 to 122: This content pertains to the methods rather than the results. Expand it and integrate it into the methods section with a dedicated paragraph.

(Our response to the comments)

Following the advice, we have added a new paragraph, 'Post-weaning rearing conditions', in the Methods section (Lines 623-643).

(Reviewer's comment)

Line 129: Add "condition" to "SI" for clarity.

(Our response to the comments)

We thank Reviewer #2 for the suggestion. We added 'mouse' or 'condition' after GH and SI throughout the revised manuscript.

(Reviewer's comment)

Lines 138 to 140: To properly assess or rule out sex differences, male and female mice should be directly compared for single-fiber amplitudes. It is not sufficient to state that the difference is reduced—this must be supported by statistical testing.

Lines 143 to 145: Sex differences should also be examined at later developmental stages, as they may emerge due to sex maturation.

Lines 148 to 153: The authors must include a comparison of male and female animals in adulthood. Psychiatric conditions such as autism spectrum disorders exhibit sex differences

at molecular and physiological levels, which may be more pronounced at specific developmental stages. The absence of differences in early development does not rule out their emergence later.

(Our response to the comments)

We thank Reviewer #2 for an interesting suggestion. Since we cannot rule out the possibility that sex differences may emerge after sexual maturation, as noted by Reviewer #2, we have added a new experiment in which adult females are reared in SI conditions. The data are presented in Supplementary Fig. 4e-h, and the results are reported in lines 133-158. We found no significant differences in the afferent numbers, SF-amplitude, and Max-amplitude between female adult GH and SI, as in the case of male adult GH and SI shown in Fig. 1e-h. However, a direct comparison of each parameter after adult SI between males and females revealed significant differences in afferent number (Lines 152-158, Fig. 1e-h, Supplementary Fig. 2b, d, 4e-h). These results suggest that the responsibility of MLF-VPM synapses to SI in females is not different from that in males in adolescence, but might be higher than in males in adulthood. The neural basis of the observed sexual differences and their involvement in behavior is a fascinating subject for future research.

(Reviewer's comment)

It is unclear how many animals were used for the experiments in Figure 1 and Extended Figure 2. The authors must clarify this.

It is not clear how many animals they have used for experiments shown in 4C-F.

(Our response to the comments)

Following the advice, the number of samples and mice was added for all experiments. Additionally, all figures were revised to display distinct symbols for each mouse, allowing the data for each mouse to be easily identified.

(Reviewer's comment)

Lines 188 to 190: The methodology used for the Von Frey test is deeply flawed. Manually restraining mice during the test introduces significant stress, which can alter sensory thresholds and render the data unreliable. Standard procedures require animals to be tested in a containment chamber after an acclimation period to minimize stress and obtain valid reflexive responses. Without these methodological corrections, the results from this dataset are invalid. The authors must revise their approach using a standardized protocol. Otherwise, the conclusions drawn from this test cannot be supported.

Lines 197 to 199: The use of the Gap Cross Test in this study is highly problematic. The experimental group had significantly higher body weight than controls during adolescence, which could have influenced the results. Since body weight directly affects motor performance and balance, the observed differences cannot be solely attributed to sensory or whisker-dependent factors but may be confounded by biomechanics. Unless body weight differences are accounted for, the conclusions drawn from this test remain unreliable. The

authors must normalize for weight differences or reconsider their interpretation of the data. This is further supported by the fact that, in adulthood, when SI mice have a body weight similar to GH mice, no difference in gap distance is observed.

To improve behavioral assessment, I strongly recommend the inclusion of a textured novel object recognition test (PMID: 39025432; PMID: 23026377) or a whisker nuisance test (PMID: 37691105; PMID: 34791077), as these are more reliable measures of whisker sensitivity.

(Our response to the comments)

We thank Reviewer #2 for their critical yet helpful comments on our behavioral assessments. As Reviewer #2 pointed out, restraining mice during the test could affect their emotional state and modulate their behavior. Thus, the deficiencies of the experimental method are explained in the result section (lines 191-194), and the data are relocated to Supplementary Fig. 6b and e. As for the result of gap-cross tests, we cannot completely rule out the possibility that the weight gain may affect the score, even the fact that the difference in the score of gap-cross tests between GH and SI disappeared after the removal of whiskers (GH vs. SI, with whiskers: $p=0.001$; without whiskers: $p=0.284$, Supplementary Fig. 6c) supported that the poor performance of SI mice depends on whisker-using sensation. Therefore, we revised our evaluation to assess the effect of whisker deprivation on the scores of gap-cross tests under each rearing condition, rather than directly comparing GH and SI. The results are presented in lines 242-261 and Supplementary Fig. 6a,c,d,f. Furthermore, according to the reviewer's suggestion, we conducted the tactile novel object recognition test (tNORT), which has been proven in previous studies to evaluate texture discrimination using whiskers (Wu et al., 2013, Behav Brain Res, DOI: 10.1016/j.bbr.2012.09.044; Kwak et al., 2020, Neuron, DOI: 10.1016/j.neuron.2020.08.013; Young et al., 2023, Nat Commun, DOI: 10.1038/s41467-023-41749-x), and the results are shown in lines 198-241 and Fig. 2. The tNORT demonstrated that mice exposed to adolescent SI showed decreased tactile discrimination compared to adolescent GH mice. The results are mentioned in lines 198-241 and Fig. 2.

(Reviewer's comment)

Lines 240 to 242: The statement here is an overstatement. Please revise it to accurately reflect the data.

(Our response to the comments)

We removed the statement.

(Reviewer's comment)

Line 242 to line 247: urinary corticosterone levels were measured at different time points in only 4 animals per condition. One of the animals for the SI condition (figure 4a) seems an outlier at day 7 (the one that shows corticosterone levels of almost 150ng/ml), also in consideration that the other 3 animals have corticosterone levels ranging between 60 and 90ng/ml. this is also crucial in consideration of the implication of the results the authors claim

as they selected the 1 week time point for successive analysis.

The very same can be found for adult SI mice on day 21 (figure 4b).

(Our response to the comments)

We thank Reviewer #2 for precise remarks. Following the advice, we increased the number of samples and confirmed the results (adolescent SI, N = 12 mice; adolescent GH, N = 14; adult GH, N = 8; adult SI, N = 8; Fig. 4a, b). The data, as noted by Reviewer #2, showed significant variability among individuals. We think the variability is partially due to differences in urinal concentration and stress levels in the hours preceding sampling. Although the variation was considerable, the value of adolescent SI was significantly higher than GH on day 7. Our consideration that elevated corticosterone in the first week after SI initiation is required for synaptic remodeling is also supported by the results of the DEX administration experiment under GH conditions (Fig. 5a), the RU486 administration experiment under SI conditions (Fig. 5b), and one-week DEX administration to adult GH mice (Supplementary Fig. 7).

(Reviewer's comment)

Line 252 to line 259: A fundamental confirmatory experiment should also assess peak amplitude in adulthood at P28. If the authors claim that corticosterone has an effect on synaptic remodelling, they should also see the effect in adult animals when corticosterone levels are higher.

(Our response to the comments)

We thank Reviewer #2 for a critical suggestion. We have conducted an experiment in which synthetic corticosterone (dexamethasone, DEX) was administered to adult mice for 1 week and are presented in Supplementary Fig. 7. The results show that DEX treatment induces an increase in the number of projectiles and a reduction of single-fiber amplitude even in adults, which supports the idea that elevated corticosterone induces synaptic remodeling. In addition, the difference in the effects of SI on synapses between adolescents and adults may be due to differences in corticosterone production by the HPA axis in response to SI (Lines 342-354).

Responses to Referee #3

(Reviewer's comment)

The manuscript by Nakayama and Miyata investigates how social isolation during adolescence impacts synaptic remodeling between medial lemniscus fibers (MLFs) and the ventral posterior medial nucleus (VPM) and subsequently altering behavior which relies on whisker somatosensory processing. The study presents intriguing evidence that remodeling occurs exclusively in adolescent social isolation and are mediated by glucocorticoid receptor (GR) activation. This remodeling disrupts the typical mono-innervation of MLFs to VPM neurons, leading to multiple innervation, weakened synaptic strength per fiber, and increased tactile hypersensitivity. While the manuscript provides compelling insights into how early-life stress can influence sensory processing and the possible underlying circuit mechanisms, several key points require further clarification. I suggest some additional experiments and data analysis (details below). There are also some clarifications about methods which need to be addressed.

(Our response to Reviewer #3)

We sincerely thank Reviewer #3 for thoroughly reviewing our manuscript and providing thoughtful, constructive suggestions and comments. In response to the comments, we revised our manuscript with additional experiments and analysis. Our responses to individual comments are as follows.

(Reviewer's comment)

Electrophysiology:

- 1. The electrophysiology recordings should be explained in more details. What is "discrete EPSC steps" (line 554)? How many sweeps were taken into account? What is the range of stimulus intensities and the delta of the gradually increased steps? How many steps are in the "3<" group? How is the number of fibers being calculated?*
- 2. It might be helpful to relabel the x-axis of the corresponding figures (e.g. Fig. 1c and g).*

(Our response to the comments)

We apologize to Reviewer #3 for our lack of explanation and thank him/her for pointing it out. We have added a method for estimating the number of fibers projecting from the MLF to the VPM neuron based on the waveform of EPSCs at different stimulus intensities in the Methods section (Lines 669-686). Also, the x-axis of the figures showing the distribution of afferent fibers was relabeled from 1 to the actual number observed.

(Reviewer's comment)

3. Amplitude plots (e.g. EDF. 1c):

- a. x-axis has a typo: change to "single fiber" instead of "single-figer"*

(Our response to the comments)

We corrected it.

(Reviewer's comment)

b. I understand that the histograms representing % cells per number of steps include all recorded cells (e.g. EDF. 1b, n=44 neurons). Then cells having monoinnervation should be ~70% of 44 = 31 neurons. Why the number of data points in "Single-fiber" in the amplitude plot is instead n=65 (EDF. 1c)? Related – how is the number of fibers being calculated and why the number of single fibers is superior to the number of recorded cells?

c. Are then cells having monoinnervation (~70% of 44 = 31 neurons) also participating in the "Max" in the amplitude plot (e.g. EDF. 1c)? If this is the case then "Max" should feature all recorded cells (n=44 neurons). Why the number of cells in "Max" does not match the total amount of cells and is instead n=43? In other data sets the "Max" features either higher or lower number of cells than the total amount of recorded neurons, please explain.

(Our response to the comments)

We apologize for the confusion caused to Reviewer #3 by the discrepancies in the sample numbers in some panels and thank him/her for pointing this out. The number of cells for the distribution of afferent numbers and the number of samples in the Max-amplitude plots are the same, and the number of samples in the single-fiber amplitude matches the total number of afferent fibers in each experimental condition. The discrepancy was solely due to our lack of confirmation. We have carefully checked and revised the data.

(Reviewer's comment)

4. Paired-pulse ratio data (lines 134-135) is missing from the figures.

(Our response to the comments)

Following this advice, we presented the PPR data as Supplementary Figs. 1 and 2.

(Reviewer's comment)

5. Clearly state the number of animals used in each condition, not just the number of cells. Given the variability observed in amplitude plots (e.g., Figure 1d, 1h), inter-animal differences could contribute to the spread of data. To address this, plots showing 'number of steps' and amplitude data for individual animals should be included to demonstrate variability across subjects.

(Our response to the comments)

Following the advice, the number of samples and mice was added for all experiments. Additionally, all figures were revised to display different symbols for each mouse, allowing the data for each mouse to be easily identified.

(Reviewer's comment)

6. The age difference between adolescent and adult social isolation groups (P56 vs P91) is a potential confound. The authors should consider testing animals that underwent adolescent

social isolation at an age matching the adult isolation group. By doing this, the authors could also: (a) Examining long-term effects of adolescent social isolation; (b) Investigating whether group housing during adulthood can reverse adolescent isolation-induced synaptic remodeling.

(Our response to the comments)

We thank Reviewer #3 for the insightful comments. In the revised manuscript, we experimentally verified whether reorganized synapses can be normalized by cohabiting adolescent 4-week or 1-week SI mice with a male age-matched GH mouse for 4 weeks (Fig. 7). The results are mentioned in lines 427-455 and discussed in lines 570-591. Shortly, we found a mild but significant normalization in the afferent number and single-fiber amplitude when adolescent 1w-SI mice were cohabited with a GH-mouse from P28 to P56 (1w-SI+4w-cohabitation (CH), P56-63 at slice experiments). However, the number of afferents in the 1w-SI+4w-CH mice was significantly higher ($p = 0.003$), and the amplitude of single fibers was significantly lower ($p = 0.005$) than in the adolescent 4w-GH mice (P49-56 at slice experiments) shown in Fig. 1a-d. These results suggest that full normalization requires additional factors beyond adolescent CH. Elucidating these factors remains a future challenge.

(Reviewer's comment)

GR involvement:

1. KO experiments: The GR-ThKO mouse does not seem to be "thalamic neuron-specific" as the authors claim. Instead, this mouse lacks GR in serotonin fibers all across the brain (raphe nuclei, multiple thalamic and cortical areas etc). The 5HTT-Cre mice used for the initial crossing does not restrict the recombination exclusively to VPM and instead expresses Cre in all 5HTT-positive fibers across brain areas. Due to the lack of such GR KO specificity, interpretation of the behavior phenotype is challenging. I'd suggest that either (1) only the slice experiments (Fig. 6e-h) are kept in this paper, and the behavior experiments (Fig.6i-k) removed due to impossibility to interpret the results (multiple brain areas possibly playing a role); or (2) changing genetic strategy with an area-specific deletion of GR in the VPM.

(Our response to the comments)

We thank Reviewer #3 for pointing out critical issues and for helpful suggestions. Regarding the gap-cross test and the von Frey test, there is concern that the observed difference in body weight after SI might influence motor performance and that restraining mice during whisker pad stimulation could alter the sensory threshold. Therefore, in the revised manuscript, that data was relocated to Supplementary Fig. 6. Instead, the tactile novel object recognition test (tNORT), which has been used as a behavioral test to evaluate texture discrimination ability using whiskers (Wu et al., 2013, Behav Brain Res, DOI: 10.1016/j.bbr.2012.09.044; Kwak et al., 2020, Neuron, DOI: 10.1016/j.neuron.2020.08.013; Young et al., 2023, Nat Commun, DOI: 10.1038/s41467-023-41749-x), was newly conducted and presented in lines 209-255 and Fig. 2.

In GR-ThKO mice isolated during adolescence, we also performed the tNORT test (Fig. 6n-r). OFT during habituation session and normal object recognition test (NORT) showed that locomotor activity and anxiety in the arena used, as well as object recognition, which depends on short-term memory, were not different in control floxed GR-mice (Fig. 6n-p, r). On the other hand, data from tNORT showed that texture discrimination ability in GR-ThKO mice improved compared to floxed-GR mice (Fig. 6q). These results are mentioned in lines 391-413. While we still cannot rule out the possibility that GR is deleted in brain regions other than the VPM and that these regions are directly or indirectly involved in tNORT and the gap-cross test (lines 417-421), we want to present these results as supporting evidence that GR in the VPM neurons plays a role in the altered tNORT performance after adolescent SI.

(Reviewer's comment)

2. What is the proposed mechanism linking neuronal GR activation in VPM and presynaptic innervation? The discussion mentions instead involvement of GR in astrocytes.

(Our response to the comments)

We appreciate Reviewer #3 for accurate remarks and constructive advice. Following his/her suggestion, we have removed the mentions about astrocyte GR and instead added a discussion of the mechanism by which neuronal GR activation induces remodeling of MLF-VPM synapses (Lines 472-501). Based on a report that increased tonic inhibition in VPM neurons leads to add afferents and reduced SF-amplitude (Nagumo et al., 2020, Cell Rep, DOI: 10.1016/j.celrep.2020.107797) and a study showing that tonic inhibition is increased in the hippocampus after SI during adolescence (P30-P60) (Serra et al., 2006, J Neurochem, DOI: 10.1111/j.1471-4159.2006.03850.x; 2008, Brain Res Rev, DOI: 10.1016/j.brainresrev.2007.06.029), we now hypothesize that tonic inhibition of VPM neurons is enhanced by SI during adolescence, resulting in synaptic remodeling downstream of the VPM neurons. We also aim to investigate the involvement of positive allosteric modulators of GABAA receptors in enhancing tonic inhibition, along with increased receptor expression and GABA release.

(Reviewer's comment)

3. Corticosterone measurements: Increase the number of animals used with at least n+2. The current conclusions are based on data from only n=4 mice.

(Our responses to the comments)

We thank Reviewer #3 for their accurate remarks. Following the advice, we increased the number of samples and confirmed the results (adolescent SI, N = 12 mice; adolescent GH, N = 14; adult GH, N = 8; adult SI, N = 8; Fig. 4a, b). The data, as noted by Reviewer #2, showed significant variability among individuals. We think the variability is partially due to differences in urinal concentration and stress levels in the hours preceding sampling. Although the variation was considerable, the value of adolescent SI was significantly higher than GH on day 7. Our consideration that elevated corticosterone in the first week after SI

initiation is required for synaptic remodeling is also supported by the results of the DEX administration experiment under GH conditions (Fig. 5a), the RU486 administration experiment under SI conditions (Fig. 5b), and one-week DEX administration to adult GH mice (Supplementary Fig. 7).

(Reviewer's comment)

4. Histology:

a. Quantification of the confocal data in EDF. 5 is missing. For the moment this figure features only example images.

b. Include in the methods part how was the analysis of VGluT2-positive puncta performed.

(Our response to the comments)

We indeed thank Reviewer #3 for pointing out this important issue. We have quantified the percentage of GR expression in neurons, astrocytes, and microglia in the thalamic VPM of WT, floxed-GR, and GR-ThKO mice, and presented the results with images (Lines 361-366 and 373-379, Fig. 6a-l, Supplementary Fig. 8). In addition, procedures for analysing the ratio of GR expression per cell type and the ratio of tdTomato-positive puncta among VGluT2-puncta were added to the Methods section (Lines 724-756).

(Reviewer's comment)

5. Corticosterone measurements: Increase the number of animals used with at least n+2. The current conclusions are based on data from only n=4 mice.

(Our response to the comments)

Please see our response to Reviewer's comment 3.

(Reviewer's comment)

Behavior:

1. To strengthen the link between synaptic remodeling and behavioral effects, additional experiments comparing MLF-VPM synaptic connectivity in animals with varying tactile sensitivity would be valuable. This could help establish a more direct relationship between the observed synaptic changes and behavioral outcomes.

(Our response to the comments)

We thank Reviewer #3 for the very valuable suggestions. We also believe that examining the correlation between behavioral test performance and synaptic status is quite significant in considering individual differences in vulnerability or tolerance to SI. This is a subject we are considering for future research, so please allow us to make the decision not to include it in this paper.

(Reviewer's comment)

Statistics:

1. I'd suggest to use nonparametric tests everywhere to avoid having data sets from similar experiments analyzed with a mix of parametric and nonparametric tests.

(Our response to the comments)

We revised to use common statistical tests with the same parameters and mentioned the statistical methods used in the figure legends and main text. We also revised the Statistics in the Methods section in accordance with the journal guidelines (Lines 839-864).

(Reviewer's comment)

2. The statistical methods section should be updated to include the two-way ANOVA. When reporting ANOVA results in the main text, the authors should use the correct format for degrees of freedom, e.g., $F(x,1) = 6.903$ and $F(x,1) = 15.549$.

(Our response to the comments)

We revised the format for reporting the results of ANOVA.

(Reviewer's comment)

3. What are the statistics used for calculating differences in % cells per number of steps (e.g. Fig. 1c and g)? This should be Fisher's test with adjustment for multiple comparisons or similar.

(Our response to the comments)

We apologize to Reviewer #3 for the confusing description. We compared the mean number of afferents between groups using a Mann-Whitney U test. In the revised manuscript, we added the mean number of afferents with the p -value. Additionally, the x-axis of the figures showing the distribution of afferent numbers was relabeled from 1 to the actual observed number.

(Reviewer's comment)

4. Mention which statistical tests were used for each data set in text and legends.

(Our response to the comments)

We added the name of the statistical test in the text and legends.

(Reviewer's comment)

5. Add N = number of animals and n = number of neurons for each data set in text and legends.

(Our response to the comments)

Following the advice, the number of samples and mice was labeled as n and N , respectively, for all experiments. Additionally, all figures were updated to display different symbols for each mouse, allowing the data from each mouse to be distinguished.

(Reviewer's comment)

Text:

1. Mention early on in the text that this study was mainly performed in male mice.

(Our response to the comments)

The use of primarily male mice is noted at the beginning of Results (Line 94), and the results of experiments using females are described in a separate section (Lines 133-158). Additionally, gender was mentioned in the Methods section (Lines 604-605) and figure legends.

(Reviewer's comment)

2. Intro: Remove and/or edit some sentences in order to manage readers expectations about this paper. For example, "psychosocial stress in adolescence impairs sensory information processing in the thalamus, thereby causing psychiatric conditions that persist into adulthood. However, no studies have directly examined this possibility at the synaptic level." (lines 86-89). This paper does not address "sensory processing", or "psychiatric conditions".

3. Intro: Define what thalamic areas the citations refer to (line 74-82). The thalamus is not a homogenous structure and different thalamic nuclei can have profound differences in connectivity and function linked to sensory processing.

(Our response to the comments)

My deepest thanks to Reviewer #3 for the supportive advice. We have revised the abstract and introduction to align with the experimental content and removed the relevant sentences. Please review the revised abstract and introduction (Lines 29-90).

(Reviewer's comment)

4. Intro: The abbreviation 'ASD' needs to be explained (line 77).

5. Intro: MLFs are first mentioned in line 93 but defined only later in line 109.

(Our response to the comments)

Abbreviations were mentioned with full names when they first appeared (MLF: line 68). The introduction has been substantially revised following comments 2 and 3 of Reviewer #3. Under the revision, the term 'autism spectrum disorder (ASD)' is now used only once in the discussion section, so the abbreviation is no longer required (Line, 497-498).

(Reviewer's comment)

6. Discussion: The discussion of adult social isolation effects should be expanded. While it doesn't cause a shift from mono- to multiple innervation, there is a significant decrease in single fiber and max EPSC amplitudes. The authors should elaborate on how this impact differs from adolescent social isolation and its potential implications.

(Our response to the comments)

We thank Reviewer #3 for the very important suggestions. As Reviewer #3 pointed out, the decrease in SF-amplitude and the increase in afferent number did not always change together; independence of their regulatory mechanisms has also been suggested during

development (Wang et al., 2011, DOI: 10.1523/JNEUROSCI.4569-11.2011; Wang et al., 2011, DOI: 10.1113/jphysiol.2011.213702). However, both the decrease in SF-amplitude and the increase in afferent number were induced by DEX administration, suggesting that they are GR-dependent (Fig. 4a, Supplementary Fig. 7). On the other hand, GR deficiency in VPM neurons suppressed the increase in afferent number induced by adolescent SI but did not rescue the attenuation of SF-amplitude (Fig. 6l, m). Taken together, we now hypothesize that different degrees of GR activation are required for the two synaptic changes caused by SI, and that the difference in thresholds may reflect the other cell types expressing the GRs involved. We have added these considerations to lines 472-501. Discussions of how these synaptic changes contribute to whisker-related behaviors in mice are provided in lines 503-538 (see also our responses to the following comments).

(Reviewer's comment)

7. Discussion: How does MLF-VPM synaptic remodeling might impact VPM output to the barrel cortex, given its critical role in tactile processing? Discussing this would help contextualize the findings within the broader somatosensory circuit.

(Our response to the comments)

We appreciate Reviewer #3 for the valuable advice, and we added a discussion on how changes at MLF-VPM synapses contribute to behavioral change in the Discussion section (Lines 503-538). Regarding our data, the SF-amplitudes were not always correlated with impairment of the tNORT or gap-cross test, as observed in adult SI mice (Fig. 1h, 2g) and GR-ThKO mice experiencing adolescent SI (Supplementary Fig. 6e, g). Thus, we believe that the decrease in SF amplitude alone does not have a sufficient effect to change these behavioral scores. Instead, our data says that the increase in afferents has a greater impact on behaviors. Morphological analysis using Krox20-Ai34D mice and VGLUT2 immunostaining has revealed that the newly formed synapses during adolescent SI bring information from the whiskers rather than ectopic regions in the brainstem (Supplementary Fig. 5). If the added synapses carry information from the whiskers that do not match the original somatotopy, the accuracy of spatial perception with the whiskers may be reduced, and the performance on tNORT and gap-cross test may be altered. We think it is necessary to test these hypotheses through morphological analysis and *in vivo* electrophysiological recording or imaging of the neural activity of VPM neurons in response to whisker stimulation in adolescent SI mice.

(Reviewer's comment)

8. Fix mislabeled figure legends.

(Our response to the comments)

We carefully checked and revised the labeling.

Responses to reviewers:

We would like to sincerely thank the reviewers for taking the time to read our responses and revised manuscript. We are especially grateful for their recognition that the revised version has improved. Furthermore, we deeply appreciate the additional questions and constructive suggestions provided at this stage. Below, we address each point in detail.

We have highlighted the changes in the main text and figure legends with light blue markers to make them easily identifiable.

Responses to Reviewer #1:

(Reviewer's comment)

The revised manuscript addressed my concerns with the original manuscript.

There was one typo in line 274 "apmlitude-> amplitude."

(Our response to Reviewer #1)

We sincerely appreciate Reviewer #1 for his/her positive evaluation of our revised manuscript and the helpful feedback. The typographical error in line 273 ("apmlitude") has been corrected to "amplitude" as suggested. Thank you again for your careful reading and constructive comments.

Responses to Reviewer #2:

(Reviewer's comment)

I thank the authors for thoroughly addressing my comments. I believe the manuscript has significantly improved and will be of interest to the journal's broad readership.

(Our response to Reviewer #2)

We sincerely appreciate Reviewer #2 for his/her acknowledgment that the revisions have sufficiently addressed the points raised, as well as his/her recognition of the improvements made to this paper and its relevance to the journal's broad readership. His/her valuable feedback has greatly helped improve the quality of our paper.

Responses to Referee #3

(Reviewer's comment)

The authors have answered to most of my comments and I do think the manuscript has significantly improved (although it would benefit from an extensive proofreading to fix typos; see comments below).

(Our response to Reviewer #3)

We sincerely thank Reviewer #3 for their constructive feedback and favorable evaluation of the revised manuscript. We appreciate the additional suggestions provided at this stage and the careful review of the remaining typographical errors. Below, we address each point in detail.

(Reviewer's comment)

However, the authors haven't answered to one of my most important concerns regarding the GR-ThKO mouse line which does not seem to be "thalamic neuron-specific" as the authors claim. If I understand correctly, this is a general and not area-specific KO mouse which lacks GR in serotonin (5HTT-positive) fibers across brain areas (raphe nuclei, multiple thalamic and cortical areas etc) and not exclusively in the VPM. This makes interpretation of the behavior results using this mouse line inconclusive. Can the authors explain in detail how the GR expression in this mouse line is specific to the thalamus?

If this is not the case, I'd again suggest to (1) change genetic strategy with an area-specific deletion of GR in the VPM, or (2) use the GR-ThKO mouse only in the slice experiments without the behavior.

(Our response to Reviewer #3)

We appreciate your thoughtful comments regarding the limitations of the 5HTT-Cre mouse line. In response, we have revised the manuscript to relocate the behavioral data to the Supplementary Figures (now Supplementary Figure 9) and clarified their exploratory nature to avoid overinterpretation as follows:

Lines 415-421: "These behavioral experiments suggest that GRs in thalamic neurons may contribute to the modulation of whisker-related behaviors following adolescent SI. While we cannot exclude the possibility that GR deletion occurred in brain regions beyond the VPM, and that such cells may directly or indirectly influence performance in the tNORT and gap-crossing tests, the present data tentatively support the notion that GRs in VPM neurons play a role in shaping whisker-related behavioral outcomes following adolescent SI."

Lines 613-615: "We generated mice with the knockout of GR ~~specifically~~ in thalamic neurons (GR-ThKO) by intercrossing GR-floxed mice with 5HTT-Cre mice."

5HTT-Cre line has previously been utilized to target thalamic neurons, though it is not strictly

VPM-specific. For example, Arakawa et al. (J Neurosci, 2014, DOI: 10.1523/JNEUROSCI.1663-14.2014) used this line to generate thalamus-specific NR1 knockout mice. They conducted behavioral analyses similar to those in our study (Gap crossing test, Object recognition test (NORT), and Texture discrimination test (tNORT). Likewise, Zhang et al. (Sci Rep, 2016, DOI: 10.1038/srep28750) employed 5HTT-Cre (Sert-Cre) mice for thalamus-specific behavioral investigations. Given the existence of these prior studies, while we acknowledge the anatomical and functional limitations of the 5HTT-Cre line, we respectfully submit that it is not appropriate to completely delete these behavioral experimental data.

Regarding Reviewer #3's suggestion to use alternative genetic approaches such as AAV or shRNA-mediated knockdown, we carefully considered these options. However, in our experimental paradigm, intracranial injections into pre-weaning pups would require maternal separation and post-injection recovery, which could confound the adolescent SI paradigm. In addition, these methods face limitations in infection efficiency, cell-type specificity, and reproducibility.

Taking these trade-offs into account, we selected the 5HTT-Cre line with full awareness of its limitations. We hope these revisions and clarifications demonstrate our commitment to scientific transparency and rigor, and respectfully address Reviewer #3's concerns regarding the genetic strategy and behavioral interpretation.

(Reviewer's comment)

Remaining points:

1. The authors discussed the influence of cohousing a male GH mouse with a male SI mouse. They showed that group housing after one week of social isolation did not fully restore MLF–VPM connectivity to GH levels. They suggest that introducing another male to an SI mouse may itself be a stressor, given that they observed attack behavior. This raises a follow-up question: what would be the cohabiting effect in female mice, who are generally less aggressive toward newly-introduced females?

(Our response to Reviewer #3)

We thank Reviewer #3 for raising such an important and stimulating point. We are also interested in the possibility that co-housing with females may offer valuable insights into the restoration of MLF–VPM synaptic connectivity remodeled by adolescent SI. However, we respectfully ask Reviewer #3 to understand that this direction will be pursued in a future study, as it falls beyond the scope of the current investigation.

(Reviewer's comment)

2-1. Statistics: : Line 849, it should be “Welch’s t-test” instead of “Wheich’s t-test.”

(Our response to Reviewer #3)

We regret that the mention of Welch's t-test was inadvertently retained during the revised submission. It has now been removed to reflect the statistical methods used in the present

study accurately. The statistical method for comparing two independent datasets has been revised as follows (lines 844-847): Group comparisons were performed using Student's t-test when data satisfied both normality (Shapiro–Wilk test) and equal variance (Brown–Forsythe test). If normality was not met, the Mann–Whitney U test was used and is referred to as 'U test' throughout the manuscript.

(Reviewer's comment)

2-2. For one-way repeated measures ANOVA, testing homogeneity of variance (e.g., Brown–Forsythe test) is not required. Instead, sphericity must be tested, and if violated, corrections such as Greenhouse–Geisser should be applied. In general, it is not accepted to replace a two-way repeated measures ANOVA with Mann–Whitney U tests. If ANOVA assumptions are strongly violated, data transformation should be attempted first.

(Our response to Reviewer #3)

We sincerely appreciate Reviewer #3's insightful comments. The point raised was highly valuable in clarifying the statistical description of our analysis, and we have revised the relevant section accordingly. Before applying the one-way repeated measures ANOVA in tNORT (Fig. 2e, g, 6q), we confirmed the assumptions of normality and sphericity using the Shapiro–Wilk test and Mauchly's test, respectively. The "equal variance (Brown–Forsythe test)" has been corrected to "sphericity (Mauchly's test)" to reflect the statistical procedure used accurately. We revised the method section as follows (lines 849-853): In the tNORT (Fig. 2e, g, Supplementary Fig. 9g), a one-way repeated measures ANOVA, followed by a post-hoc test of Holm-Sidak multiple comparisons, was used after confirming the assumptions of normality and sphericity using the Shapiro–Wilk test and Mauchly's test, respectively.

In this study, the CORT data did not meet the assumption of normality needed for two-way repeated measures ANOVA. Therefore, we attempted to build a general linear mixed model (GLMM) using JMP's Generalized Linear Mixed Model platform. However, due to problems with model convergence and uncertainty in specifying the variance structure, reliable estimation proved to be difficult. Next, we considered using nonparametric two-way repeated measures analysis (e.g., Aligned Rank Transform (ART), Rank-based Analysis of Factorial Designs (rankFD)). Still, these methods were not available in the statistical software we had access to, and could not be practically implemented. As an exploratory step, we performed separate Mann–Whitney U tests to evaluate the effect of rearing condition at each postnatal age. Although ignoring the repeated measures structure raises concerns about increased false positives, we believe the reliability of the significance observed 7 days after the start of GH/SI is reasonably supported by results from independent experiments of one-week administration of DEX and RU in Fig. 5. Moving forward, we plan to use nonparametric methods capable of adequately handling repeated measures designs, like rankFD, for more rigorous statistical testing.

3. *Supplementary Figure 1a & 1b: Please provide a description of the color code used.*

(Our response to Reviewer #3)

The color labels were initially intended to indicate which waveform on the right side of the plot corresponded to each point in the main graph. However, as pointed out by Reviewer #3, we agree that these labels may have caused confusion rather than clarification. Therefore, we have removed the color labels from the figure and sincerely appreciate the reviewer's helpful suggestion.

Text & typos:

(Reviewer's comment)

1. *The sentence between lines 78–79 describes the study's findings, which fits better in the following paragraph (lines 80–90). Please either revise this sentence or move it to the next paragraph to improve the flow.*

(Our response to Reviewer #3)

We appreciate Reviewer #3's insightful suggestions. To improve the logical flow, we have integrated the relevant section into the final sentence of the following section (lines 86-89):
Our study demonstrates that adolescence is a crucial period when vulnerability to social isolation increases, and stress-induced GR overactivation can alter thalamic neural circuits and disrupt sensory processing.

(Reviewer's comment)

2. *Post-weaning rearing condition: The first sentence (Lines 624–625) is somewhat confusing; please rephrase for clarity.*

(Our response to Reviewer #3)

We thank you for Reviewer #3's advice. We have rephrased the relevant section as follows (lines 623-625):
In most experiments, male mice were weaned at 21 days postnatal and randomly assigned to two housing conditions: the group housing (GH) group and the social isolation (SI) group. Female mice were used in Supplementary Fig. 2c, d, and 4.

(Reviewer's comment)

3. *Line 117: The N for the number of GH afferents is missing. Please recheck the cell number for the SI group; currently it is written as n = 3.*

(Our response to Reviewer #3)

We revised as follows (line 116): afferent number: GH, 1.49 ± 0.11 , $n = 37$, $N = 3$; SI, 2.14 ± 0.15 , $n = 37$, $N = 3$, $p < 0.001$ by *U*-test.

(Reviewer's comments)

4. *Line 123: Please replace "SH" with "SI."*

5. *Line 152: Typo — should be "after" instead of "afater."*

6. *Line 295: Typo — should be "difference" instead of "dirrence."*

7. Line 304: Typo — should be “recordings” instead of “recoedings.”

8. Line 528: Typo — should be “form” instead of “from.”

9. Line 897: Typo — should be “arena” instead of “area.”

(Our response to Reviewer #3)

We appreciate Reviewer #3 for thoroughly reviewing the manuscript.

4. We have corrected “SH” to “SI” (line 123).

5. We have corrected “*afater*” to “after” (line 151).

6. We have corrected “*dirrerence*” to “difference” (line 294).

7. We have corrected “*recoedings*” to “recordings” (line 303).

8. We have rephrased “the adolescent SI induced synapses *from* new afferents” to “the adolescent SI induced synapse formation by new afferents” (lines 528-529).

9. We have rephrased “dashed *area*” to “The area surrounded by a dashed line” (line 892).